# Optimized tight binding between the S1 segment and KCNE3 is required for the constitutively open nature of the KCNQ1-KCNE3 channel complex

**Go Kasuya\*, Koichi Nakajo\***

Division of Integrative Physiology, Department of Physiology, Jichi Medical University, Shimotsuke, Japan

**Abstract** Tetrameric voltage-gated K⁺ channels have four identical voltage sensor domains, and they regulate channel gating. KCNQ1 (Kv7.1) is a voltage-gated K⁺ channel, and its auxiliary subunit KCNE proteins dramatically regulate its gating. For example, KCNE3 makes KCNQ1 a constitutively open channel at physiological voltages by affecting the voltage sensor movement. However, how KCNE proteins regulate the voltage sensor domain is largely unknown. In this study, by utilizing the KCNQ1-KCNE3-calmodulin complex structure, we thoroughly surveyed amino acid residues on KCNE3 and the S1 segment of the KCNQ1 voltage sensor facing each other. By changing the side-chain bulkiness of these interacting amino acid residues (volume scanning), we found that the distance between the S1 segment and KCNE3 is elaborately optimized to achieve the constitutive activity. In addition, we identified two pairs of KCNQ1 and KCNE3 mutants that partially restored constitutive activity by co-expression. Our work suggests that tight binding of the S1 segment and KCNE3 is crucial for controlling the voltage sensor domains.

## Editor's evaluation

This study of physiologically important K⁺ channel complexes is expected to be important to electro-physiologists and biophysicists. This structure-motivated mutagenesis and biophysical study provide compelling evidence that a structural interface between a K⁺ channel and a class of modulatory subunits is a critical functional interface that determines the efficacy of the modulatory subunits.

**\*For correspondence:**
gokasuya@jichi.ac.jp (GK);
knakajo@jichi.ac.jp (KN)

**Competing interest:** The authors declare that no competing interests exist.

## Introduction

KCNQ1 (Kv7.1) is a voltage-gated K⁺ channel. Its gating behavior depends mainly on its auxiliary subunit KCNE proteins (*Wang et al., 2020*). There are five KCNE genes in the human genome, and all of them are known to modify KCNQ1 channel gating behavior, at least in *Xenopus* oocytes or mammalian cell lines (*Bendahhou et al., 2005*). Therefore, the physiological functions of the KCNQ1 channel are determined by the type of KCNE proteins that are co-expressed in a tissue. The most well-studied example is the cardiac KCNQ1-KCNE1 channel, which underlies the slow cardiac delayed-rectifier K⁺ current ($I_{Ks}$) (*Barhanin et al., 1996*; *Sanguinetti et al., 1996*; *Takumi et al., 1988*). Another example is the KCNQ1-KCNE3 channel, a constitutively open channel expressed in epithelial cells in the trachea, stomach, and intestine. This channel complex couples with some ion transporters to facilitate ion transport by recycling K⁺ (*Abbott, 2016*; *Grahammer et al., 2001*; *Preston et al., 2010*; *Schroeder et al., 2000*).

The mechanisms by which KCNE proteins modify KCNQ1 channel gating behavior have been a central question of this ion channel. Since KCNQ1 is a classic *shaker*-type tetrameric voltage-gated K⁺ channel, it has four independent voltage sensor domains (VSDs), one from each α subunit (*Long et al., 2005*). Each VSD consists of four transmembrane segments, S1–S4. Each S4 segment bears several positively charged amino acid residues. When the membrane potential is depolarized, each S4 segment moves upward (toward the extracellular side). That upward movement eventually triggers pore opening (*Jensen et al., 2012*; *Larsson et al., 1996*; *Mannuzzu et al., 1996*). Therefore, the S4 segment is considered to be an essential part of voltage sensing (*Aggarwal and MacKinnon, 1996*; *Catterall, 1988*; *Fedida and Hesketh, 2001*; *Liman et al., 1991*; *Logothetis et al., 1992*; *Noda et al., 1984*; *Papazian et al., 1991*). As in the *Shaker* K⁺ channel, the S4 segment of the KCNQ1 channel also moves upward with depolarization, as proved by scanning cysteine accessibility mutagenesis (*Nakajo and Kubo, 2007*; *Rocheleau and Kobertz, 2008*) and voltage-clamp fluorometry (VCF; *Barro-Soria et al., 2014*; *Nakajo and Kubo, 2014*; *Osteen et al., 2012*; *Osteen et al., 2010*). In those studies, the presence of KCNE proteins was found to affect the voltage sensor movement. These results suggest that KCNE proteins stabilize a specific state during the voltage sensor transition (*Barro-Soria et al., 2017*; *Barro-Soria et al., 2015*; *Barro-Soria et al., 2014*; *Nakajo, 2019*). There should be at least three positions in the VSDs of KCNQ1: 'down,' 'intermediate,' and 'up' (*Hou et al., 2017*; *Taylor et al., 2020*). KCNE1 stabilizes the intermediate position of VSDs along with a direct interaction of the pore domain and inhibits opening of the pore domain (*Taylor et al., 2020*). In contrast, KCNE3 may stabilize the intermediate or up position of VSDs and indirectly stabilize the channel's open state (*Barro-Soria et al., 2017*; *Barro-Soria et al., 2015*; *Taylor et al., 2020*). However, it remains unknown how KCNE proteins stabilize VSDs at a specific position.

Early studies by Melman et al. revealed 'the triplet' of amino acid residues in the middle of the transmembrane segment ('FTL': F57-T58-L59 in KCNE1 and 'TVG': T71-V72-G73 in KCNE3) as structural determinants of KCNE modulation properties (*Melman et al., 2002*; *Melman et al., 2001*). Exchanging the triplet (or one of the three amino acid residues) between KCNE1 and KCNE3 could introduce the other's modulation properties, at least partially. For example, introducing 'FTL' into KCNE3 transforms it into a KCNE1-like protein. Therefore, it has been long considered that 'the triplet' is a functional interaction site between KCNQ1 and KCNE proteins. Possible interaction sites of the KCNQ1 side have also been explored and are believed to be located between the pore domain and the VSD (*Chung et al., 2009*; *Kang et al., 2008*; *Nakajo and Kubo, 2007*; *Van Horn et al., 2011*; *Xu et al., 2008*). By utilizing the KCNQ1 ortholog from ascidian *Ciona intestinalis*, which lacks KCNE genes, we previously found that F127 and F130 of the S1 segment are required for KCNE3 to make KCNQ1 a constitutively open channel (*Nakajo et al., 2011*). A recent computational model and the cryo-EM structure of the KCNQ1-KCNE3-calmodulin (CaM) complex clearly showed that KCNE3 interacts with the S1 segment and the pore domain (*Kroncke et al., 2016*; *Sun and MacKinnon, 2020*). However, the mechanism by which KCNE3 retains the KCNQ1 VSD at a specific position is still not clearly understood.

In this work, by taking advantage of the cryo-EM structure of the KCNQ1-KCNE3-CaM complex (*Sun and MacKinnon, 2020*), we created a series of the S1 segment and KCNE3 mutants to change the bulkiness of the S1 segment and KCNE3 interface, and we found that the interaction between the S1 segment and KCNE3 is elaborately optimized to achieve the constitutive activity. In addition, we identified two pairs of the S1 segment and KCNE3 mutants that partially restored constitutive activity by co-expression. Our work suggests tight binding of the S1 segment and KCNE3 is crucial for controlling the VSDs.

## Results

### The side-chain volumes of amino acid residues in the S1 segment of KCNQ1 facing KCNE3 are optimized for channel modulation

The cryo-EM structure of the KCNQ1-KCNE3-CaM complex revealed that KCNE3 interacts with the S1 segment of KCNQ1 in the transmembrane segment (*Sun and MacKinnon, 2020*). In the structure, the side chains of seven amino acid residues in the S1 segment of KCNQ1 (F123, F127, F130, L134, I138, L142, and I145) face lining those of six amino acid residues of the KCNE3 transmembrane segment (S57, I61, M65, A69, G73, and I76; *Figure 1*). We previously reported that F127 and F130

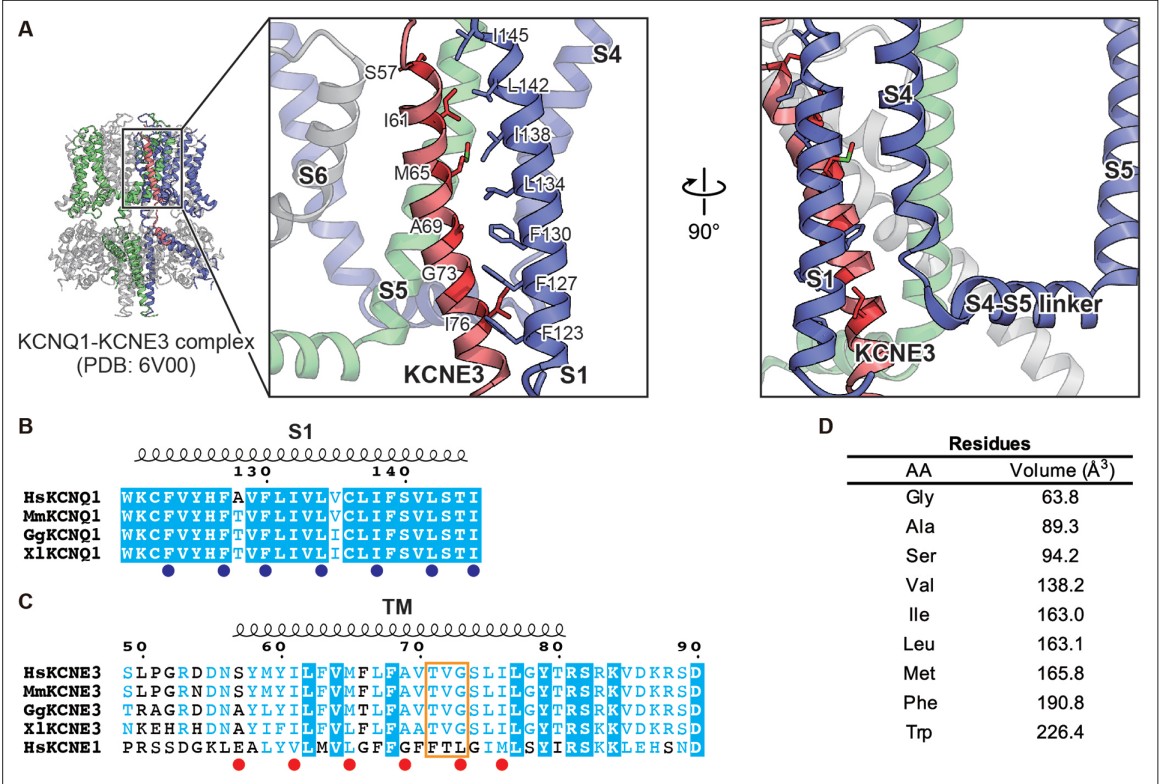

**Figure 1.** Key residues involved in the interaction between KCNQ1 and KCNE3. (**A**) Close-up view of the interface between KCNQ1 and KCNE3 in the KCNQ1-KCNE3-CaM complex structure (PDB: 6V00). Three KCNQ1 subunits are colored in blue, green, and gray. A KCNE3 subunit is colored in red. The residues involved in the KCNQ1-KCNE3 interaction are depicted by stick models. The molecular graphics were illustrated with CueMol (http://www.cuemol.org/). (**B and C**) Sequence alignment around the S1 segment of KCNQ1 (**B**) and the TM segments of KCNE3 and KCNE1 (**C**). Amino acid sequences were aligned using Clustal Omega and are shown using ESPript3 (***Robert and Gouet, 2014***). KCNQ1 residues focused on in this work are highlighted with blue dots. KCNE3 residues focused on in this work and 'the triplet' (***Melman et al., 2002***; ***Melman et al., 2001***) are highlighted with red dots and an orange square, respectively. For sequence alignment, human KCNQ1 (HsKCNQ1, NCBI Accession Number: NP_000209), mouse KCNQ1 (MmKCNQ1, NP_032460), chicken KCNQ1 (GgKCNQ1, XP_421022), *Xenopus* KCNQ1 (XlKCNQ1, XP_018111887), human KCNE3 (HsKCNE3, NP_005463), mouse KCNE3 (MmKCNE3, NP_001177798), chicken KCNE3 (GgKCNE3, XP_003640673), *Xenopus* KCNE3 (XlKCNE3 NP_001082346), and human KCNE1 (HsKCNE1, NP_000210) were used. (**D**) The sizes of amino acid residues focused on in this work. The numbers are from ***Tsai et al., 1999***.

are required to make KCNQ1-KCNE3 a constitutively open channel (***Nakajo and Kubo, 2011***). Therefore, we hypothesized that the interactions between the S1 segment of KCNQ1 and KCNE3 might be crucial for stabilizing the open states. To confirm the functional roles of these amino acid residues, we first created and tested small alanine and large tryptophan-substituted mutants of the S1 segment of KCNQ1. If the alanine- and/or tryptophan-substituted mutants changed the functional output induced by the KCNQ1-KCNE3 interaction (i.e. did not show the constitutive activity), we also created and tested intermediate-sized hydrophobic residues valine-, leucine-, and phenylalanine-substituted mutants. When expressed alone, most of the S1 segment mutants showed conductance-voltage (G-V) curves similar to that of KCNQ1 WT (***Figure 2—figure supplements 1–7***). We then tested how these mutations introduced to the S1 segment affect the modulation by KCNE3. When co-expressed with KCNE3 WT, KCNQ1 WT shifted the G-V curve in the far-negative direction, becoming a constitutively open channel for the physiological membrane potential range (***Figure 2A, B, G and L***; black traces and black G-V curves). Because of that, the relative conductance of KCNQ1 WT-KCNE3 WT at –100 mV ($G_{-100mV}/G_{max}$) was 10-times higher (KCNQ1 WT, 0.08±0.01; KCNQ1 WT-KCNE3 WT, 0.80±0.02; n=10 for each; ***Figure 2 M-S***; black bars). In contrast, when co-expressed with KCNE3 WT, the S1 segment mutants yielded various G-V curves and $G_{-100mV}/G_{max}$ values. Since one of the unique features of the KCNE3 modulation is constitutive KCNQ1 activity even in the hyperpolarized voltage range, we then mainly evaluated the effect of each mutation on KCNE3 modulation by the $G_{-100mV}/G_{max}$ value for comparison. However, other parameters, such as the midpoint of the G–V curve ($V_{1/2}$) or

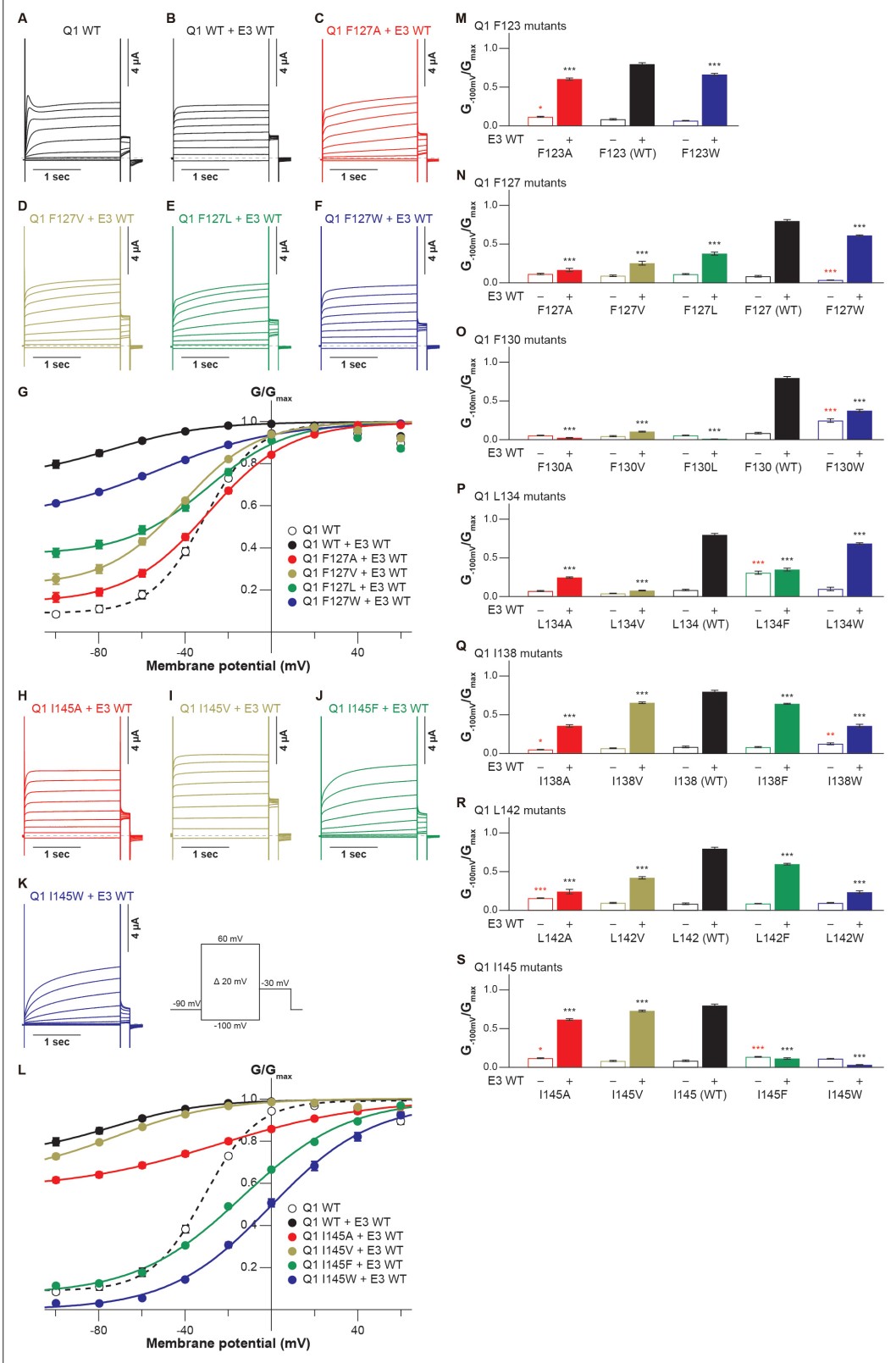

**Figure 2.** Functional effects of KCNQ1 S1 mutants on KCNQ1 modulation by KCNE3. (**A–G**) Representative current traces (**A–F**) and conductance-voltage (G-V) relationships (**G**) of KCNQ1 WT with or without KCNE3 WT as well as the F127 mutants with KCNE3 WT. (**H–L**) Representative current traces (**H–K**) and G-V relationships (**L**) of the I145 mutants with KCNE3 WT. (**M–S**) Ratios of conductance at −100 mV ($G_{-100mV}$) and maximum conductance ($G_{max}$)

*Figure 2 continued on next page*

*Figure 2 continued*

of KCNQ1 F123 (**M**), F127 (**N**), F130 (**O**), L134 (**P**), I138 (**Q**), L142 (**R**), and I145 (**S**) mutants with (filled bars) or without (open bars) KCNE3 WT. Error bars indicate ± SEM for n=10 in (**G, L, and M–S**).

The online version of this article includes the following source data and figure supplement(s) for figure 2:

**Source data 1.** Summary of the electrophysiological properties of KCNQ1 WT and mutants with or without KCNE3 WT.

**Source data 2.** Excel file with numerical electrophysiology data used for *Figure 2*.

**Figure supplement 1.** Current traces and conductance-voltage (G-V) relationships of KCNQ1 F123 mutants.

**Figure supplement 2.** Current traces and conductance-voltage (G-V) relationships of KCNQ1 F127 mutants.

**Figure supplement 3.** Current traces and conductance-voltage (G-V) relationships of KCNQ1 F130 mutants.

**Figure supplement 4.** Current traces and conductance-voltage (G-V) relationships of KCNQ1 L134 mutants.

**Figure supplement 5.** Current traces and conductance-voltage (G-V) relationships of KCNQ1 I138 mutants.

**Figure supplement 6.** Current traces and conductance-volltage (G-V) relationships of KCNQ1 L142 mutants.

**Figure supplement 7.** Current traces and conductance-voltage (G-V) relationships of KCNQ1 I145 mutants.

---

effective charge ($z$), were evaluated when possible (*Figure 2—source data 1*, *Figure 3—source data 1*, and *Figure 4—source data 1*). Both of the F123 mutants (F123A and F123W) shifted the G-V curves in the far-negative direction and kept the $G_{-100mV}/G_{max}$ values over 0.5 (F123A, 0.61±0.01; F123W, 0.66±0.01; n=10 for each; *Figure 2M* and *Figure 2—figure supplement 1*), suggesting that the F123 residue did not have a large impact on the KCNE3 modulation. Among the F127 mutants (F127A, F127V, F127L, and F127W), F127A, F127V, and F127L mutants failed to shift the G-V curves in the far-negative direction and showed smaller $G_{-100mV}/G_{max}$ values depending on the side-chain size (F127A, 0.17±0.02; F127V, 0.25±0.02; F127L, 0.38±0.02; n=10 for each; *Figure 2C–G and N* and *Figure 2—figure supplement 2*). It seemed that the modulation depended on the side-chain volume: the more different the size was, the more significant the change of functional output induced by the KCNQ1-KCNE3 interaction was (*Figure 2N*). The F130 mutants (F130A, F130V, F130L, and F130W) failed to shift the G-V curves in the far-negative direction or even positively shifted the G-V curves. They showed substantially reduced $G_{-100mV}/G_{max}$ values (F130A, 0.02±0.00; F130V, 0.10±0.01; F130L, 0.01±0.00; F130W, 0.37±0.02; n=10 for each; *Figure 2O* and *Figure 2—figure supplement 3*). Among the L134 mutants (L134A, L134V, L134F, and L134W), the L134A, L134V, and L134F mutants failed to shift the G-V curves in the far-negative direction and reduced the $G_{-100mV}/G_{max}$ values (L134A, 0.25±0.01; L134V, 0.08±0.01; L134F, 0.35±0.02; n=10 for each; *Figure 2P* and *Figure 2—figure supplement 4*). All the I138 mutants (I138A, I138V, I138F, and I138W) showed relatively mild attenuation in the G-V shift and $G_{-100mV}/G_{max}$ values (I138A, 0.36±0.02; I138V, 0.66±0.01; I138F, 0.64±0.01; I138W, 0.36±0.02; n=10 for each; *Figure 2Q* and *Figure 2—figure supplement 5*). The I138 mutants showed an explicit size dependency in the KCNE3 modulation with wild-type isoleucine having the largest impact on the KCNE3 modulation, as in the case of the F127 mutants (*Figure 2N*). The L142 mutants (L142A, L142V, L142F, and L142W) failed to shift the G-V curves in the negative direction and mildly reduced the $G_{-100mV}/G_{max}$ values (L142A, 0.24±0.03; L142V, 0.42±0.02; L142F, 0.60±0.01; KCNQ1 L142W-KCNE3 WT, 0.23±0.02; n=10 for each; *Figure 2R* and *Figure 2—figure supplement 6*). Again, the L142 mutants showed an explicit size dependency in the KCNE3 modulation with wild-type leucine, as in the case of the F127 and I138 mutants (*Figure 2N, Q and R*). Among the I145 mutants (I145A, I145V, I145F, and I145W), smaller amino acid substitutions (I145A and I145V) showed similar functional outputs to that of WT. In contrast, the I145F and I145W mutants positively shifted the G-V curves and reduced the $G_{-100mV}/G_{max}$ values as the side-chain size was increased (I145F, 0.11±0.01; I145W, 0.03±0.00; n=10 for each; *Figure 2H–L and S* and *Figure 2—figure supplement 7*).

In summary, when mutated, six of seven tested amino acid residues (F127, F130, L134, I138, L142, and I145) in the S1 segment changed the functional output induced by the KCNQ1-KCNE3 interaction. Five of them (F127, F130, I138, L142, and I145) showed some side-chain volume dependency for the modulation by KCNE3: KCNQ1 WT showed the highest modulation effect by KCNE3, and the modulation effects by KCNE3 changed gradually if introduced mutations at the S1 segment became more different from WT in terms of amino acid size. Therefore, a series of side-chain volumes facing

KCNE3 in the S1 segment is tightly optimized and vital for proper gating modulation of KCNQ1 induced by KCNE3.

## The side-chain volumes of amino acid residues of KCNE3 facing the KCNQ1 S1 segment are also optimized for channel modulation

Next, we assessed the functional role of the six amino acid residues of KCNE3 facing the S1 segment (*Figure 1*). As in the case of evaluating the S1 segment, we created various mutants of these residues and co-expressed them with KCNQ1 WT. Throughout the experiments, 10 ng RNA of KCNQ1 was co-injected with 1 ng RNA of KCNE3 into oocytes since 1 ng RNA of KCNE3 was sufficient to fully modulate KCNQ1 currents (*Figure 3—figure supplement 1*). In the S57 mutants (S57G, S57A, S57V, S57L, S57F, and S57W), the $G_{-100mV}/G_{max}$ values were more gradually reduced when introduced mutations became more different from WT, as seen in some S1 mutants (S57G, 0.32±0.02; S57A, 0.66±0.02; S57V, 0.47±0.01; S57L, 0.20±0.01; S57F, 0.10±0.01; S57W, 0.10±0.01; n=10 for each; *Figure 3A–G and N*). Among the I61 mutants (I61G, I61A, I61V, I61F, and I61W), the I61V mutant reduced its $G_{-100mV}/G_{max}$ value to 0.57±0.02 (n=10), but the value was still relatively high. The other four I61 mutants shifted the G-V curves in the positive direction and greatly reduced their $G_{-100mV}/G_{max}$ values (I61G, 0.04±0.01; I61A, 0.03±0.01; I61F, 0.04±0.00; I61W, 0.11±0.01; n=10 for each; *Figure 3O* and *Figure 3—figure supplement 2*). The M65 mutants (M65G, M65A, M65V, M65L, M65F, and M65W) showed similar side-chain volume dependency as seen in the S57 mutants and some S1 mutants. They showed less shifted G-V curves and reduced the $G_{-100mV}/G_{max}$ values depending on how different the side-chain size was from that of WT, except M65V, which showed a significant change of functional output despite having a size similar to that of WT (M65G, 0.22±0.01; M65A, 0.48±0.01; M65V, 0.16±0.02; M65L, 0.63±0.03; M65F, 0.36±0.02; M65W, 0.07±0.01; n=10 for each; *Figure 3P* and *Figure 3—figure supplement 3*). Among the A69 mutants (A69G, A69V, A69L, A69F, and A69W), only the smaller A69G mutant showed a similar functional output to that of WT at this position. A69G shifted the G-V curve in the far-negative direction and kept the $G_{-100mV}/G_{max}$ value over 0.5 (0.57±0.03, n=10), while the other four A69 mutants shifted the G-V curve in the positive direction and showed substantially smaller $G_{-100mV}/G_{max}$ values (A69V, 0.19±0.02; A69L, 0.17±0.01; A69F, 0.11±0.01; A69W, 0.04±0.01, n=10 for each; *Figure 3Q* and *Figure 3—figure supplement 4*). Only the small amino acid alanine showed a similar functional output to that of WT again in the G73 mutants (G73A, G73V, G73L, G73F, and G73W; *Figure 3H–M and R*). The G73A mutant shifted the G-V curve in the far-negative direction and kept the $G_{-100mV}/G_{max}$ value over 0.5 (0.77±0.01, n=10). The other four G73 mutants mildly shifted the G-V curve in the negative direction and reduced the $G_{-100mV}/G_{max}$ values to variable extents (G73V, 0.22±0.02; G73L, 0.35±0.02; G73F, 0.43±0.02; G73W, 0.24±0.01; n=10 for each; *Figure 3H–M and R*). All of the I76 mutants (I76G, I76A, I76V, I76F, and I76W) showed reduced $G_{-100mV}/G_{max}$ values, depending on the side-chain volume (I76G, 0.06±0.01, n=10; I76A, 0.14±0.02, n=10; I76V, 0.43±0.02, n=10; I76F, 0.11±0.01, n=10; I76W, 0.16±0.01; n=10 for each; *Figure 3S* and *Figure 3—figure supplement 5*).

In summary, when mutated, all of the tested amino acid residues in KCNE3 showed some changes in functional output induced by the KCNQ1-KCNE3 interaction. Furthermore, KCNE3 WT showed the highest ability to modulate the KCNQ1 gating among all of the tested amino acid residues. In most cases, the abilities to modulate the KCNQ1 gating changed more significantly if the size of introduced mutations differed more from WT. As in the case of the S1 segment (*Figure 2* and *Figure 2—figure supplements 1–7*), these results demonstrate that a series of side-chain volumes facing the S1 segment in KCNE3 is tightly optimized for proper gating modulation of KCNQ1 induced by KCNE3.

## Functional restoration of the S1 mutants by the KCNE3 mutants

We then examined whether the KCNE3 mutations could restore the functional output of the KCNQ1-KCNE3 interaction distorted by the S1 segment mutations. As guided by the KCNQ1-KCNE3-CaM complex structure (*Sun and MacKinnon, 2020*), we tested the idea with five candidate pairs positioned to the same layer in the structure (F127[Q1]-G73[E3], F130[Q1]-A69[E3], I138[Q1]-M65[E3], L142[Q1]-I61[E3], and I145[Q1]-S57[E3]; *Figure 1*). In the F127(Q1)-G73(E3) pairs, we chose the KCNQ1 F127A mutant since it showed the largest change of functional output among the F127 mutants. Among the KCNE3 G73 mutants, the F127A-G73L pair showed the largest restoration of the $G_{-100mV}/G_{max}$ value (0.67±0.01, n=10) as compared to that of the F127A-KCNE3 WT pair (0.17±0.02, n=10; *Figure 4A–G*).

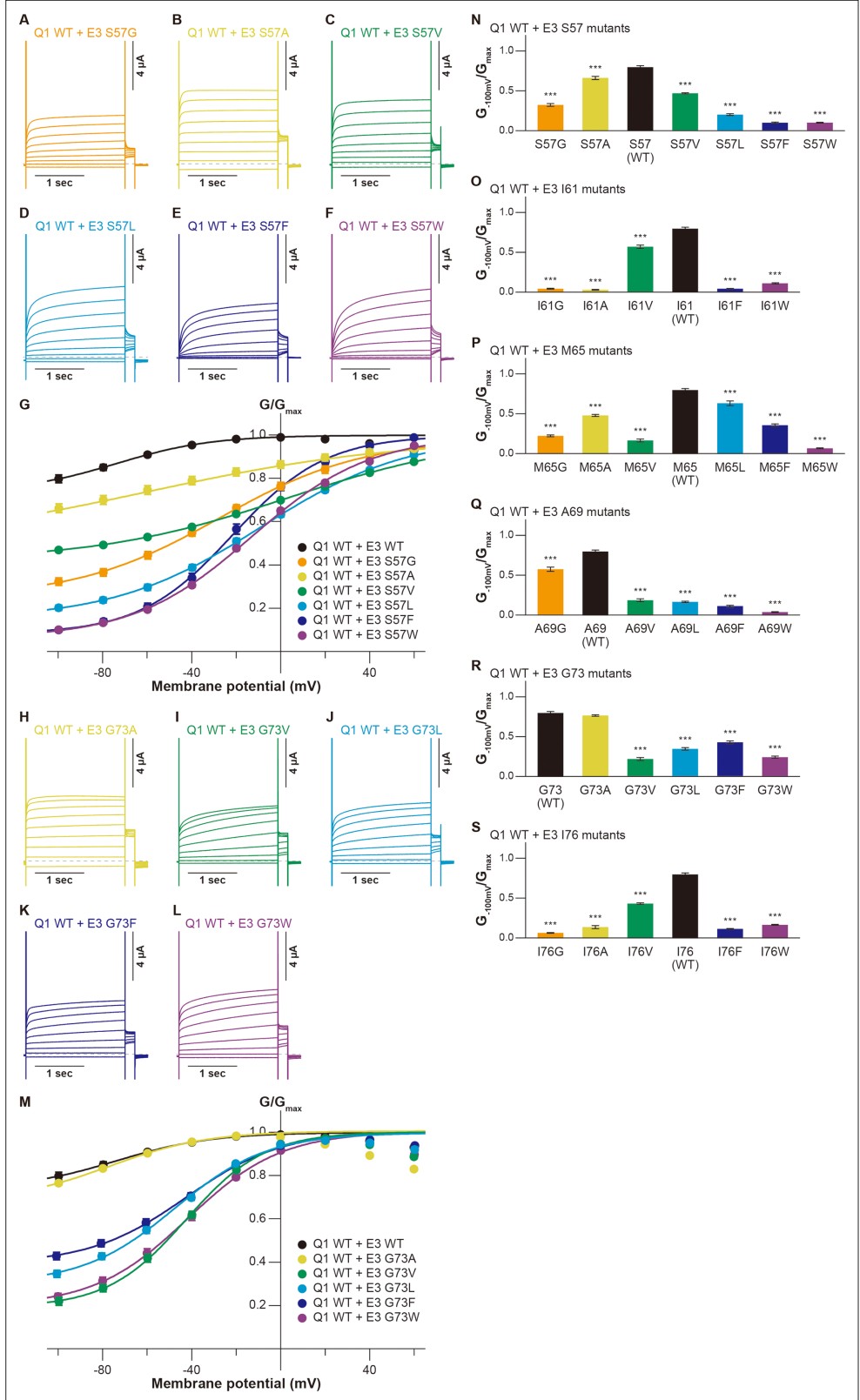

**Figure 3.** Functional effects of KCNE3 mutants on KCNQ1 modulation by KCNE3. (**A–G**) Representative current traces (**A–F**) and G-V relationships (**G**) of KCNQ1 WT with the KCNE3 S57 mutants. (**H–M**) Representative current traces (**H–L**) and G-V relationships (**M**) of KCNQ1 WT with the KCNE3 G73 mutants. (**N–S**) Ratios of conductance

*Figure 3 continued on next page*

*Figure 3 continued*

at −100 mV ($G_{-100mV}$) and maximum conductance ($G_{max}$) of KCNQ1 WT with KCNE3 S57 (**N**), I61 (**O**), M65 (**P**), A69 (**Q**), G73 (**R**), and I76 (**S**) mutants. Error bars indicate ± SEM for n=10 in (**G, M,N–S**).

The online version of this article includes the following source data and figure supplement(s) for figure 3:

**Source data 1.** Summary of the electrophysiological properties of KCNQ1 WT with KCNE3 mutants.

**Source data 2.** Excel file with numerical electrophysiology data used for *Figure 3*.

**Figure supplement 1.** Dose-dependent modulation of KCNQ1 by decreasing amounts of KCNE3 WT and mutants.

**Figure supplement 2.** Current traces and conductance-voltage (G-V) relationships of KCNQ1 WT with the KCNE3 I61 mutants.

**Figure supplement 3.** Current traces and conductance-voltage (G-V) relationships of KCNQ1 WT with the KCNE3 M65 mutants.

**Figure supplement 4.** Current traces and conductance-voltage (G-V) relationships of KCNQ1 WT with the KCNE3 A69 mutants.

**Figure supplement 5.** Current traces and conductance-voltage (G-V) relationships of KCNQ1 WT with the KCNE3 I76 mutants.

Other G73 mutants, G73V (0.36±0.02, n=10), G73F (0.45±0.02, n=10), and G73W (0.38±0.01, n=10), also mildly but significantly restored the modulation (*Figure 4A–G*). In the F130(Q1)-A69(E3) pairs, we examined whether KCNQ1 F130A and F130V mutants were restored by a series of the KCNE3 A69 mutants since they showed large changes in functional output among the KCNQ1 F130 mutants. Although KCNE3 A69G, A69F, and A69W mutants slightly increased the $G_{-100mV}/G_{max}$ value of the KCNQ1 F130A mutant (A69G, 0.10±0.01; A69F, 0.05±0.00; A69W, 0.10±0.01; KCNE3 WT, 0.02±0.00; n=10 for each), most of the KCNE3 A69 mutants failed to restore the KCNQ1 F130A and F130V mutants (*Figure 4—figure supplement 1*). In the I138(Q1)-M65(E3) pairs, we examined whether KCNQ1 I138A and I138W mutants were restored by a series of the KCNE3 M65 mutants since they showed large changes in functional output among the KCNQ1 I138 mutants. In the KCNQ1 I138A-KCNE3 M65 pairs, the KCNQ1 I138A-KCNE3 M65L and KCNQ1 I138A-KCNE3 M65F pairs showed some restoration of the $G_{-100mV}/G_{max}$ values (KCNQ1 I138A-KCNE3 M65L, 0.52±0.02, n=10; KCNQ1 I138A-KCNE3 M65F, 0.53±0.01, n=10) as compared to that of the KCNQ1 I138A-KCNE3 WT pair (0.36±0.02, n=10; *Figure 4—figure supplement 2A–H*). The KCNQ1 I138W-KCNE3 M65L pair also showed mild restoration of the $G_{-100mV}/G_{max}$ value (0.46±0.02, n=10; *Figure 4—figure supplement 2I–P*), although it is not clear why both I138A and I138W were restored by the same M65L mutant. In the L142(Q1)-I61(E3) pairs, we examined whether KCNQ1 L142A and L142W mutants were restored by a series of the KCNE3 I61 mutants. However, none of the pairs were restored (*Figure 4—figure supplement 3*). In the I145(Q1)-S57(E3) pairs, we examined whether KCNQ1 I145F and I145W mutants were restored by a series of the KCNE3 S57 mutants. Although the KCNQ1 I145W mutant was slightly restored by the KCNE3 S57 mutants, no KCNE3 S57 mutants showed a $G_{-100mV}/G_{max}$ value higher than 0.18 (*Figure 4—figure supplement 4*). In contrast, KCNQ1 I145F was successfully restored by KCNE3 S57G and S57A mutants with smaller residues than WT ($G_{-100mV}/G_{max}$ values of I145F-S57G, 0.43±0.01; I145F-S57A, 0.61±0.02; n=10 for each) as compared to that of the I145F-KCNE3 WT pair (0.11±0.01, n=10; *Figure 4H–O*).

These results suggest that at least two of the five pairs of residues tested (F127[Q1]-G73[E3] and I145[Q1]- S57[E3]) closely interact with each other. This is the strong evidence that the specific interaction between the S1 segment and KCNE3 is important.

## The KCNE3 mutants restored voltage sensor movement of the S1 mutants

Previous electrophysiological studies demonstrated that KCNE3 influences the voltage sensor movement, especially the S4 segment movement (*Barro-Soria et al., 2017*; *Barro-Soria et al., 2015*; *Nakajo and Kubo, 2007*; *Rocheleau and Kobertz, 2008*). We next performed VCF to monitor the S4 segment to investigate whether the mutations of the S1 segment affect voltage sensor movement and whether it is restored by the identified pairs of mutants (F127A[Q1]-G73L[E3] and I145F[Q1]-S57A[E3]). The KCNQ1 construct for VCF (KCNQ1 C214A/G219C; this construct hereafter being referred to as

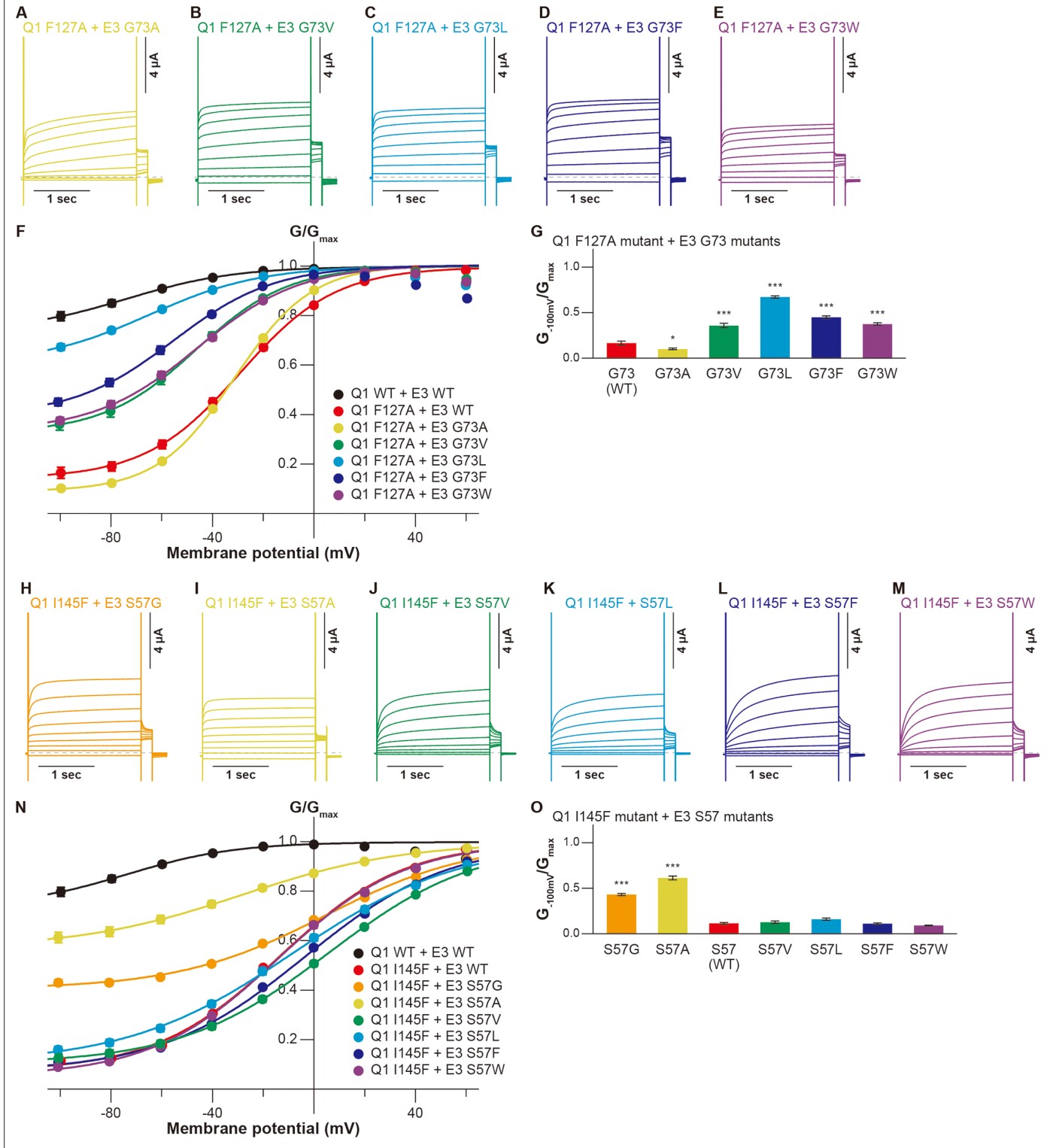

**Figure 4.** Functional restoration of KCNQ1 mutants by KCNE3 mutants. (**A–G**) Representative current traces (**A–E**), conductance-voltage (G-V) relationships (**F**), and ratios of conductance at −100 mV ($G_{−100mV}$) and maximum conductance ($G_{max}$) (**G**) of the KCNQ1 F127A mutant with the KCNE3 G73 mutants. (**H–O**) Representative current traces (**H–M**), G-V relationships (**N**), and ratios of conductance at −100 mV ($G_{−100mV}$) and maximum conductance ($G_{max}$) (**O**) of the KCNQ1 I145F mutant with the KCNE3 S57 mutants. Error bars indicate ± SEM for n=10 in (**F, G, N, and O**).

The online version of this article includes the following source data and figure supplement(s) for figure 4:

**Source data 1.** Summary of the electrophysiological properties of KCNQ1 mutants with KCNE3 mutants.

*Figure 4 continued on next page*

*Figure 4 continued*

**Source data 2.** Excel file with numerical electrophysiology data used for *Figure 4*.

**Figure supplement 1.** Current traces, conductance-voltage (G-V) relationships, and ratios of conductance of the KCNQ1 F130 mutants with the KCNE3 A69 mutants.

**Figure supplement 2.** Current traces, conductance-voltage (G-V) relationships, and ratios of conductance of the KCNQ1 I138 mutants with the KCNE3 M65 mutants.

**Figure supplement 3.** Current traces, conductance-voltage (G-V) relationships, and ratios of conductance of the KCNQ1 L142 mutants with the KCNE3 I61 mutants.

**Figure supplement 4.** Current traces, conductance-voltage (G-V) relationships, and ratios of conductance of the KCNQ1 I145W mutant with the KCNE3 S57 mutants.

'KCNQ1$_{vcf}$ WT') was labeled at the introduced cysteine residue (G219C) by Alexa Fluor 488 maleimide (*Osteen et al., 2012*; *Osteen et al., 2010*). Currents and fluorescence changes were recorded in response to voltage steps (from 60 to –160 mV) from a holding potential of –100 mV (*Figure 5* inset). KCNQ1$_{vcf}$ WT alone showed a fluorescence-voltage (F-V) relationship that mostly fitted to a single Boltzmann function and closely overlapped with its G-V curve (*Figure 5—figure supplement 1A, D and E*). In contrast, KCNQ1$_{vcf}$ WT co-expressed with KCNE3 WT showed a split F-V relationship. Its fluorescence changes were observed in the far negative and positive voltages, while they were very small and remained almost unchanged within the voltage range between 0 and –100 mV (*Figure 5A, F and G*). Consequently, the F-V relationship of KCNQ1$_{vcf}$ WT-KCNE3 WT does not fit to a single Boltzmann function but fits to a double Boltzmann function (*Figure 5A, F and G*), which is consistent with the results of a previous VCF study (*Taylor et al., 2020*). These results suggest that most of the S4 segments of the channels are at the down position and move to the upper position with depolarization in KCNQ1$_{vcf}$ WT alone and that a substantial number of S4 segments are in the intermediate position and move either with depolarization or deep hyperpolarization in KCNQ1$_{vcf}$ WT-KCNE3 WT. We then assessed the F-V relationships of the KCNQ1$_{vcf}$ F127A mutant co-expressed with the KCNE3 WT or G73L mutant as well as those of the KCNQ1$_{vcf}$ I145F mutant co-expressed with the KCNE3 WT or S57A mutant. The KCNQ1$_{vcf}$ F127A and F145F mutants alone showed G-V and F-V relationships similar to those of KCNQ1$_{vcf}$ WT alone (*Figure 5—figure supplement 1B–E*). The KCNQ1$_{vcf}$ F127A mutant co-expressed with KCNE3 WT showed an F-V relationship that still fitted to a double Boltzmann function but lost a plateau phase observed in KCNQ1$_{vcf}$ WT-KCNE3 WT (*Figure 5B and G*), resulting in a shift of the half-activation voltage in the first fluorescence component ($V_{1/2[F1]}$), which seems to be correlated to pore opening/closure, toward the positive direction (–77.1±3.0 mV, n=5) as compared to that of KCNQ1$_{vcf}$ WT-KCNE3 WT (–141.7±10.8 mV, n=5). In contrast, the KCNQ1$_{vcf}$ F127A mutant co-expressed with the KCNE3 G73L mutant showed an F-V relationship that better fitted to a double Boltzmann function and shifted its $V_{1/2(F1)}$ toward a more negative direction (< –160 mV, n=5) than the KCNQ1$_{vcf}$ F127A mutant co-expressed with KCNE3 WT (*Figure 5C and G*). These results suggest that the number of S4 segments in the intermediate position were increased in the KCNQ1$_{vcf}$ F127A mutant co-expressed with the KCNE3 G73L mutant.

A similar tendency was observed in the KCNQ1$_{vcf}$ I145F mutant co-expressed with the KCNE3 WT or S57A mutant (*Figure 5D, E, I*). The KCNQ1$_{vcf}$ I145F mutant co-expressed with KCNE3 WT showed an F-V relationship that still fitted to a double Boltzmann function but lost a plateau phase and shifted its $V_{1/2(F1)}$ toward the positive direction (–121.2±11.6 mV, n=5). In contrast, the KCNQ1$_{vcf}$ I145F mutant co-expressed with the KCNE3 S57A mutant showed an F-V relationship that better fitted to a double Boltzmann function and shifted its $V_{1/2(F1)}$ toward a more negative direction (< –160 mV, n=5). These results suggest that the number of S4 segments in the intermediate position were increased in the KCNQ1$_{vcf}$ I145F mutant co-expressed with the KCNE3 S57A mutant. This is the strong evidence that it is an interaction between the S1 segment and KCNE3 that is important for modulating for VSD movement.

## The side-chain volumes of amino acid residues in the S1 segment of KCNQ1 are important for channel modulation by KCNE1

We finally examined whether the side-chain volume of amino acid residues in the S1 segment also influences the KCNQ1 modulation by KCNE1. We co-expressed a series of the KCNQ1 F127 mutants

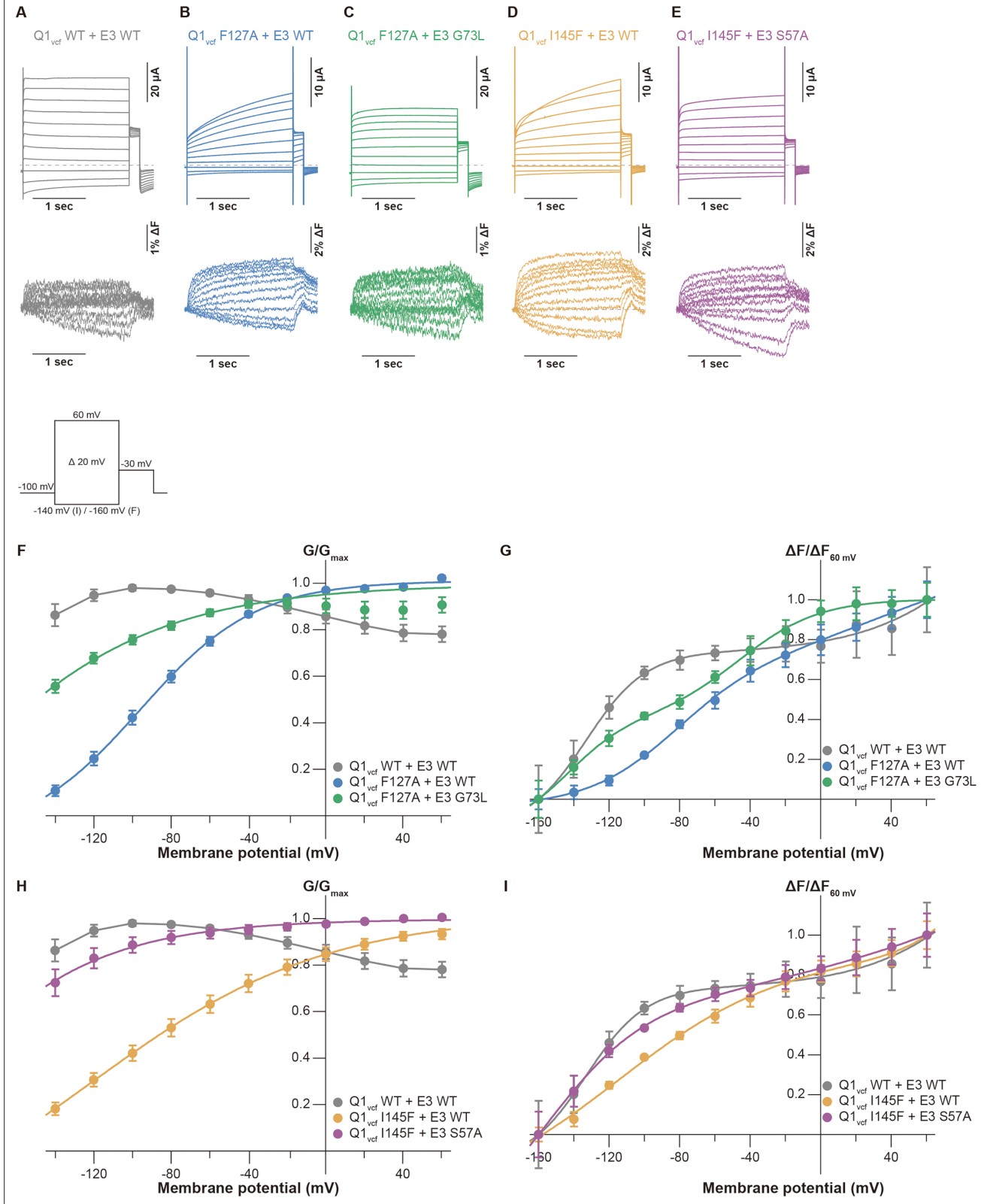

**Figure 5.** Conductance-voltage (G-V) and fluorescence-voltage (F-V) relationships for KCNQ1 mutants with KCNE3 mutants. (**A–E**) Ionic currents (upper row) and fluorescence traces (lower row) of KCNQ1$_{vcf}$ WT-KCNE3 WT (**A**), KCNQ1$_{vcf}$ F127A-KCNE3 WT (**B**), KCNQ1$_{vcf}$ F127A-KCNE3 G73L (**C**), KCNQ1$_{vcf}$ I145F-KCNE3 WT (**D**), and KCNQ1$_{vcf}$ I145F-KCNE3 S57A (**E**). (**F–I**) G-V (**F and H**) and F-V (**G and I**) relationships of KCNQ1$_{vcf}$ WT-KCNE3 WT, KCNQ1$_{vcf}$

*Figure 5 continued on next page*

*Figure 5 continued*

F127A-KCNE3 WT, KCNQ1$_{vcf}$ F127A-KCNE3 G73L, KCNQ1$_{vcf}$ I145F-KCNE3 WT, and KCNQ1$_{vcf}$ I145F-KCNE3 S57A. Error bars indicate ± SEM for n=5 in (**F–I**).

The online version of this article includes the following source data and figure supplement(s) for figure 5:

**Source data 1.** Summary of the electrophysiological properties of KCNQ1$_{vcf}$ WT and mutants with or without KCNE3 mutants.

**Source data 2.** Summary of the properties of KCNQ1$_{vcf}$ WT and mutants with or without KCNE3 mutants acquired from voltage-clamp fluorometry (VCF) recordings.

**Source data 3.** Excel file with numerical electrophysiology data used for *Figure 5*.

**Figure supplement 1.** Conductance-voltage (G-V) and fluorescence-voltage (F-V) relationships for KCNQ1 mutants.

(F127A, F127V, F127L, and F127W) with KCNE1 WT. All the F127 mutants shifted the G-V curve in the positive direction by co-expression of KCNE1. Interestingly, the induced shift depended on the side-chain volume again: the more different side-chain sizes from WT (F127) were, the larger $V_{1/2}$ shifted (*Figure 6* and *Figure 6—source data 1*; *Figure 6—source data 2*). The G-V curve could not fit to a Boltzmann equation properly in the case of F127A mutant with KCNE1, although it was apparent that $V_{1/2}$ is larger than +60 mV (*Figure 6K* inset). While KCNE1 WT and KCNE3 WT shift the G-V curve of KCNQ1 WT in opposing directions, both modulation effects seemed similarly dependent on the tight interaction of the S1 segment. However, since we examined only the F127 mutants in this study, further experiments will be needed to reveal the possible roles of the S1 segment in the modulation by KCNE1.

## Discussion

In this work, we conducted site-directed mutational analyses using two-electrode voltage clamp (TEVC) and VCF by changing the side-chain bulkiness of these interacting amino acid residues (volume scanning), inspired by the recently determined cryo-EM structures of the KCNQ1-KCNE3-CaM complex (*Sun and MacKinnon, 2020*). We found that the hydrophobic interface between the S1 segment and KCNE3 is a key component for the channel modulation by KCNE3, which prevents the S4 segment of the VSD from going to the down position at resting membrane potential. Previous studies demonstrated that 'the triplet' of amino acid residues in the middle of the transmembrane segment ('FTL' for KCNE1 and 'TVG' for KCNE3) is a structural determinant of KCNE modulation properties (*Barro-Soria et al., 2017*; *Melman et al., 2002*; *Melman et al., 2001*). However, why the triplets are important for conferring specific gating properties onto KCNQ1 is still not well understood even though the KCNQ1-KCNE3-CaM complex structure revealed that the triplets are located deep inside the membrane and interact with the S1 and S4 segments of KCNQ1 (*Sun and MacKinnon, 2020*; *Figure 1A*). Besides 'the triplet', our current work showed that a broader range of amino acid residues (S57, I61, M65, A69, G73, and I76), which forms five helical turns in total in the middle of the transmembrane segment of KCNE3, was involved in the interactions between KCNE3 and the S1 segment and was required for maintenance of constitutive activity of the KCNQ1-KCNE3 channel (*Figure 3* and *Figure 3—figure supplements 1–5*).

Another previous study by Barro-Soria et al. suggested that negatively charged KCNE3 residues (D54 and D55) electrostatically interact with the S4 segment of KCNQ1 to induce constitutive activity of the KCNQ1-KCNE3 channel (*Barro-Soria et al., 2015*). In this work, we did not investigate these negatively charged KCNE3 residues since they are far away from the S1 and S4 segments and do not directly interact with them in the KCNQ1-KCNE3-CaM complex structure (*Sun and MacKinnon, 2020*). Further mutational analyses can provide insights into the mechanism of how these residues are involved in gating modulation of KCNQ1 induced by KCNE3.

For the KCNQ1 side, we previously demonstrated that two phenylalanine residues, F127 and F130, in the S1 segment are important for the channel modulation by KCNE3, but how these mutations affect the channel modulation has been unknown (*Nakajo et al., 2011*). The cryo-EM KCNQ1-KCNE3-CaM structure (PDB: 6V00) clearly shows that F127 and F130 of the S1 segment face KCNE3 (G73 and I76). We therefore hypothesized that the interaction of these amino acid residues is essential for the KCNE3 function and found that besides F127 and F130, a broader range of amino acid residues (F127, F130, L134, I138, L142, and I145), which forms five helical turns in total in the middle

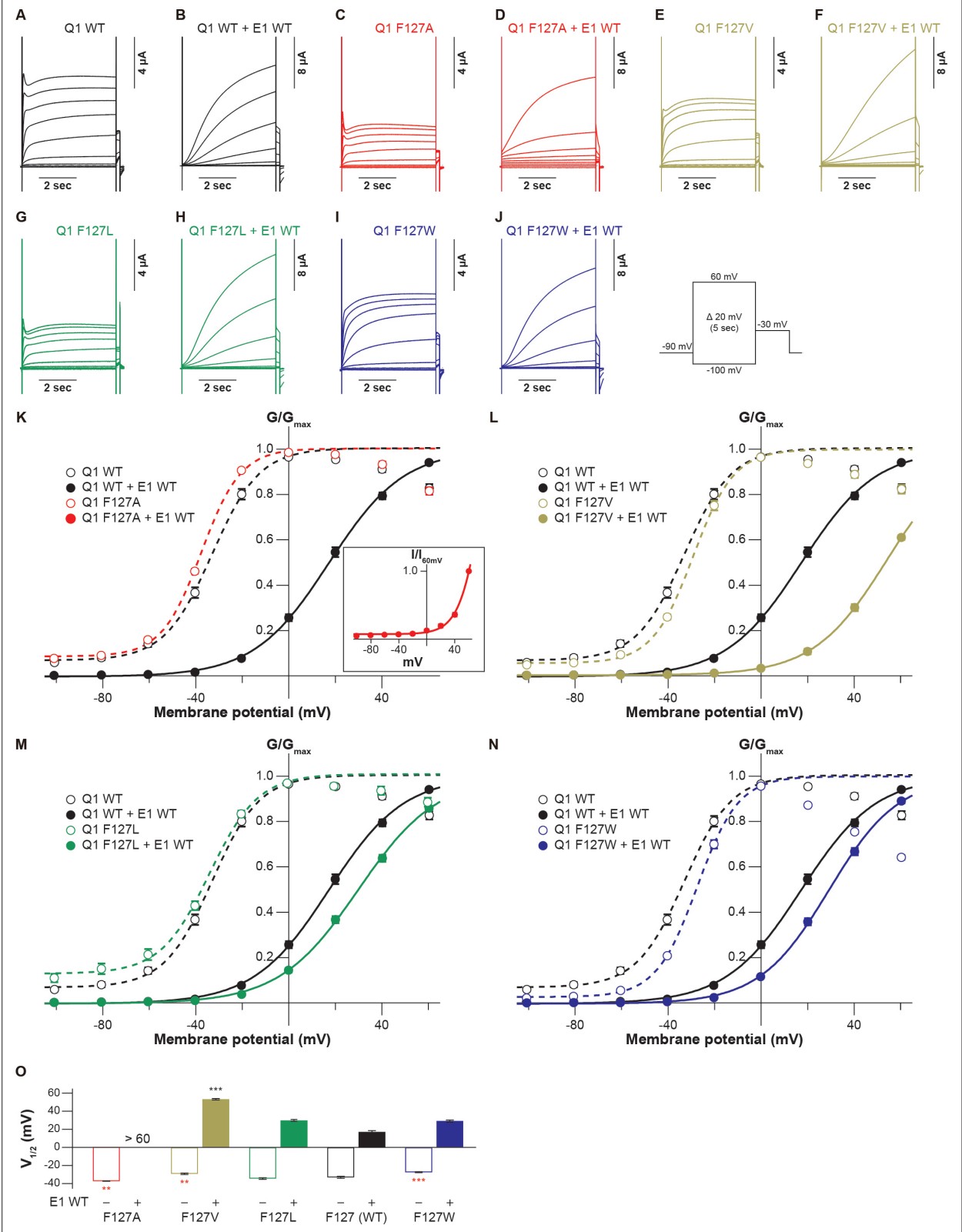

**Figure 6.** Functional effects of KCNQ1 F127 mutants on KCNQ1 modulation by KCNE1. (**A–N**) Representative current traces (**A–J**) and conductance-voltage (G-V) relationships (**K–N**) of KCNQ1 WT and F127 mutants with or without KCNE1 WT. In panel (**A**), the current-voltage (I-V) relationship of KCNQ1 F127A with KCNE1 normalized by tail current amplitudes at 60 mV ($I_{60mV}$) is shown as an inset since its G-V curve shifted in the far-positive direction and could not fit to a single Boltzmann equation properly. (**O**) The half-activation voltage of KCNQ1 WT and F127 mutants with (filled bars) or

*Figure 6 continued on next page*

*Figure 6 continued*

without (open bars) KCNE1 WT. '>60' means over 60 mV, as the G-V curve could not properly fit to a single Boltzmann equation. Error bars indicate ± SEM for n=5 in (**I–M**).

The online version of this article includes the following source data for figure 6:

**Source data 1.** Summary of the electrophysiological properties of KCNQ1 F127 mutants with KCNE1 WT.

**Source data 2.** Excel file with numerical electrophysiology data used for *Figure 6*.

of the transmembrane segment of the S1 segment, was important for the channel modulation by KCNE3. Furthermore, in most cases, the introduction of larger and smaller amino acid residues at the hydrophobic interface between the S1 segment and KCNE3 disrupted the KCNE3 function to modulate KCNQ1 gating (*Figure 2* and *Figure 2—figure supplements 1–7*). This suggests that the side-chain volume of the amino acid residues in the S1 segment and KCNE3 is optimized for proper gating modulation of KCNQ1 induced by KCNE3 (*Figure 2*, *Figure 2—figure supplements 1–7*, *Figure 3*, and *Figure 3—figure supplements 2–5*). The restorations of the KCNQ1 F127A mutant by the KCNE3 G73L mutant and the KCNQ1 I145F mutant by the KCNE3 S57A mutant further support this idea (*Figure 4*). In contrast, three of the five KCNQ1 residues tested (F130, I138, and L142) were not successfully restored by KCNE3 mutants (*Figure 4—figure supplements 1–4*). They could have even more prominent roles in the KCNE3 modulation than just the tight hydrophobic interaction tested in this study. Further analyses are needed to understand the functions of these residues in gating modulation by KCNE3.

According to previous VCF studies (*Barro-Soria et al., 2017*; *Barro-Soria et al., 2015*; *Taylor et al., 2020*), we further performed VCF analysis to investigate how the S1 segment and KCNE3 interaction influences the voltage sensor movement. Previous VCF analysis (*Taylor et al., 2020*) revealed that KCNQ1$_{vcf}$ WT co-expressed with KCNE3 WT showed two components (F1 and F2) in the F-V relationship (*Figure 5A, F and G*). In addition, KCNQ1$_{vcf}$ F127A-KCNE3 WT and KCNQ1$_{vcf}$ I145F-KCNE3 WT pairs showed positively shifted $V_{1/2(F1)}$ and diminished the plateau phase observed in the KCNQ1$_{vcf}$ WT-KCNE3 WT pair, while KCNQ1$_{vcf}$ F127A-KCNE3 G73L and KCNQ1$_{vcf}$ I145F-KCNE3 S57A pairs partially restored $V_{1/2(F1)}$ and the plateau phase (*Figure 5*). These results suggest that the tight interaction between the S1 segment and KCNE3 is required to keep the S4 segment in the intermediate position at resting membrane potential. Disrupting the interaction between the S1 segment and KCNE3 by mutation allows the S4 segment to move down by hyperpolarization.

How the tight interaction between the S1 segment and KCNE3 identified in the current study that affects the S4 segment is still up in the air. In the KCNQ1-KCNE3-CaM complex structure, the S4 segment directly interacts with KCNE3 and the S1 segment through its lower part (*Sun and MacKinnon, 2020*). Therefore, it is reasonable to speculate that the tight interaction between the S1 segment and KCNE3 directly affects the S4 movement upon membrane potential change. However, further mutational analyses are needed to understand how the tight interaction between the S1 segment and KCNE3 changes the S4 movement.

It is interesting whether the tight interaction we saw in the KCNQ1-KCNE3 channel also plays a role in the KCNQ1-KCNE1 channel. We examined a series of KCNQ1 F127 mutants co-expressed with KCNE1 and found that the mutant effects also showed size dependency, as in the case of KCNE3 (*Figure 6*). It is noteworthy, though, that the F127 mutants strengthened the positive shift of the G-V curve in the KCNQ1-KCNE1 channel, while the same mutants reduced the constitutive activity in the KCNQ1-KCNE3 channel. As previously revealed by some VCF experiments, KCNE1 and KCNE3 stabilize the intermediate state of the VSD (*Barro-Soria et al., 2015*; *Barro-Soria et al., 2014*; *Osteen et al., 2010*; *Taylor et al., 2020*). One possible interpretation of the KCNE1 results could be that the mutations of F127 might destabilize the intermediate state (or stabilize the closed/down state), as seen in KCNE3 (*Figure 5*). Therefore, the S1 segment might have a similar role in both KCNE1 and KCNE3 to assist in stabilizing the intermediate state. However, further experiments will be needed to find out the possible role of the S1 segment in the KCNE1 modulation.

In conclusion, our results demonstrate that tight interaction between the S1 segment of the KCNQ1 channel and KCNE3 is required for retaining the VSD in the intermediate position, probably by preventing the S4 segment from going to the down position, thereby keeping the KCNQ1-KCNE3 channels constitutively active.

# Materials and methods

## Expression in *Xenopus laevis* oocytes

The human *KCNQ1* (NCBI Accession Number NP_000209.2; WT and mutants), human *KCNE1* (HsKCNE1, NP_000210), and mouse *Kcne3* (NP_001177798; WT and mutants) cDNAs were inserted into the pGEMHE expression vector (*Liman et al., 1992*). The cRNAs were transcribed using mMESSAGE mMACHINE T7 Transcription Kits (Thermo Fisher Scientific, AM1344). Oocytes were surgically removed from female *X. laevis* frogs anesthetized in water containing 0.1% tricaine (Sigma-Aldrich, E10521) for 15–30 min. The oocytes were treated with collagenase (Sigma-Aldrich, C0130) for 6–7 hr at room temperature to remove the follicular cell layer. Defolliculated oocytes of similar sizes at stage V or VI were selected, microinjected with 50 nl of cRNA solution (10 ng for KCNQ1 and 1 ng for KCNE3) using a NANOJECT II (Drummond Scientific Co.), and incubated until use at 18°C in Barth's solution (88 mM NaCl, 1 mM KCl, 2.4 mM NaHCO$_3$, 10 mM HEPES, 0.3 mM Ca[NO$_3$]$_2$, 0.41 mM CaCl$_2$, and 0.82 mM MgSO$_4$, pH 7.6) supplemented with 0.1% penicillin-streptomycin solution (Sigma-Aldrich, P4333). All experiments were approved by the Animal Care Committee of Jichi Medical University (Japan) under protocol no. 18027–03 and were performed according to guidelines.

## Two-electrode voltage clamp

cRNA-injected oocytes were incubated for 1–3 days. Ionic currents were recorded with a two-electrode voltage clamp using an OC-725C amplifier (Warner Instruments) at room temperature. The bath chamber was perfused with Ca$^{2+}$-free ND96 solution (96 mM NaCl, 2 mM KCl, 2.8 mM MgCl$_2$, and 5 mM HEPES, pH 7.6) supplemented with 100 μM LaCl$_3$ to block endogenous hyperpolarization-activated currents (*Osteen et al., 2010*). The microelectrodes were drawn from borosilicate glass capillaries (Harvard Apparatus, GC150TF-10) using a P-1000 micropipette puller (Sutter Instrument) to a resistance of 0.2–1.0 MΩ and filled with 3 M KCl. Currents were elicited from the holding potential of –90 mV to steps ranging from –100 to +60 mV in +20 mV steps each for 2 s with 7.5 s intervals for the KCNQ1-KCNE3 complex analyses and for 5 s with 15 s intervals for the KCNQ1-KCNE1 complex analyses. Oocytes with a holding current larger than –0.2 μA at –90 mV were excluded from the analysis. Generation of voltage-clamp protocols and data acquisition were performed using a Digidata 1550 interface (Molecular Devices) controlled by pCLAMP 10.7 software (Molecular Devices). Data were sampled at 10 kHz and filtered at 1 kHz.

## Voltage dependence analysis

G-V relationships were taken from tail current amplitude at –30 mV fitted using pCLAMP 10.7 software (Molecular Devices) to a single Boltzmann equation:

$$G = G_{min} + (G_{max} - G_{min})/(1 + e^{-zF[V - V_{1/2}]RT}),$$

where G$_{max}$ and G$_{min}$ are the maximum and minimum tail current amplitudes, respectively, z is the effective charge, V$_{1/2}$ is the half-activation voltage, T is the temperature in degrees Kelvin, F is Faraday's constant, and R is the gas constant. G/G$_{max}$, which is the normalized tail current amplitude, was plotted against membrane potential for presentation of the G-V relationships.

## Voltage-clamp fluorometry

Sample preparation, data acquisition, and data analysis were performed similarly as described previously (*Nakajo and Kubo, 2014*). cRNA-injected oocytes were incubated for 4–5 days, labeled for 30 min with 5 μM Alexa Fluor 488 C$_5$ maleimide (Thermo Fisher Scientific, A10254) in high potassium KD98 solution (98 mM KCl, 1.8 mM CaCl$_2$, 1 mM MgCl$_2$, and 5 mM HEPES, pH 7.6; *Nakajo and Kubo, 2014*; *Osteen et al., 2010*), and washed with Ca$^{2+}$-free ND96 solution to remove unreacted Alexa probes. The bath chamber was filled with Ca$^{2+}$-free ND96 solution supplemented with 100 μM LaCl$_3$. The microelectrodes were drawn from borosilicate glass capillaries (Harvard Apparatus, GC150TF-15). Currents were elicited from the holding potential of –100 mV to steps ranging from +60 to –160 mV in –20 mV steps each for 2 s with 10 s intervals. Oocytes with a holding current larger than –0.3 μA at –100 mV were excluded from the analysis. Generation of voltage-clamp protocols and data acquisition was performed using a Digidata 1440 A interface (Molecular Devices) controlled by pCLAMP 10.7 software (Molecular Devices). Data were sampled at 10 kHz and filtered at 1 kHz.

Fluorescence recordings were performed with a macro zoom microscope MVX10 (Olympus) equipped with a 2× objective lens (MVPLAPO 2XC, NA = 0.5, Olympus), 2× magnification changer (MVX-CA2X, Olympus), GFP filter cube (U-MGFPHQ/XL, Olympus), and an XLED1 LED light source with a BDX (450–495 nm) LED module (Excelitas Technologies). Fluorescence signals were obtained by using a photomultiplier (H10722-110; Hamamatsu Photonics) and digitized at 1 kHz through Digidata1440, filtered at 50 Hz, and recorded using pClamp10 simultaneously with ionic currents. The shutter for the excitation was open during the recording, which induced a continuous decrease of fluorescence due to photobleaching. Therefore, we calculated the bleaching rate for each experiment using the baseline levels of the initial 1100 ms before test pulses of each trace and compensated the fluorescence traces by subtracting the bleached component calculated from each trace's bleaching rate (R), assuming that the fluorescence was linearly decreased. Arithmetic operations were performed with Clampfit software from pClamp10.

(Compensated trace) = (recorded trace) × (1 – [R × (time)]) (*Nakajo and Kubo, 2014*), where (time) is the time value of the point given by Clampfit. We then normalized the fluorescence traces by setting each baseline level to 1.

## VCF analysis

F-V relationships were taken from the fluorescence change from the baseline (ΔF) plotted against membrane potential. ΔF values were then normalized by $\Delta F_{60mV}$ for the normalized F-V relationships shown in *Figure 5* and *Figure 5—figure supplement 1*. For KCNQ1 alone, F-V relationships were fitted using Igor Pro software (WaveMatrices Co.) to a single Boltzmann equation:

$$F = F_{min} + (F_{max} - F_{min})/(1 + e^{-zF[V-V_{1/2}]RT}),$$

where $F_{min}$ and $F_{max}$ are the maximum and baseline fluorescence components, z is the effective charge for the fluorescence component, $V_{1/2}$ is the half-activation voltage for the fluorescence component, T is the temperature in degrees Kelvin, F is Faraday's constant, and R is the gas constant. For KCNQ1 co-expressed with KCNE3, F-V relationships were fitted using Igor Pro software (WaveMatrices Co.) to a double Boltzmann equation:

$$F = F_{min} + (F_1 - F_{min})/(1 + e^{-z1F[V-V_{1/2}(F1)]RT}) + (F_2 - F_1)/(1 + e^{-z2F[V-V_{1/2}(F2)]RT})$$

where $F_1$, $F_2$, and $F_{min}$ are the first, second, and baseline fluorescence components, z1 and z2 are the effective charges for each fluorescence component, $V_{1/2(F1)}$ and $V_{1/2(F2)}$ are the half-activation voltage for each fluorescence component, T is the temperature in degrees Kelvin, F is Faraday's constant, and R is the gas constant.

## Statistical analysis

The data were expressed as means ± SEM. Statistical analysis was performed with Student's *t*-test and one-way ANOVA with Dunnett's test for single and multiple comparisons, respectively, with EZR software (*Kanda, 2013*), and significance was assigned at $p < 0.05$ (*$p < 0.05$, **$p < 0.01$, and ***$p < 0.001$).

## Acknowledgements

We thank Dr. Yuichiro Fujiwara (Kagawa University) and the Nakajo laboratory members for their valuable discussions. This work was supported by the Japan Society for the Promotion of Science (JSPS) KAKENHI (Grant Nos. 19K23833 and 20H03200 to GK and 17K08552 and 21K06786 to KN), by JICHI MEDICAL UNIVERSITY YOUNG INVESTIGATOR AWARD to GK, and by The Salt Science Research Foundation (Grant No. 2219) to GK.

## Additional information

### Funding

| Funder | Grant reference number | Author |
|---|---|---|
| Japan Society for the Promotion of Science | 19K23833 | Go Kasuya |
| Japan Society for the Promotion of Science | 20H03200 | Go Kasuya |
| Japan Society for the Promotion of Science | 17K08552 | Koichi Nakajo |
| Japan Society for the Promotion of Science | 21K06786 | Koichi Nakajo |
| Salt Science Research Foundation | 2219 | Go Kasuya |
| JICHI MEDICAL UNIVERSITY YOUNG INVESTIGATOR AWARD | | Go Kasuya |

The funders had no role in study design, data collection and interpretation, or the decision to submit the work for publication.

### Author contributions

Go Kasuya, Conceptualization, Data curation, Formal analysis, Funding acquisition, Validation, Investigation, Visualization, Methodology, Writing - original draft, Project administration, Writing - review and editing; Koichi Nakajo, Conceptualization, Data curation, Formal analysis, Supervision, Funding acquisition, Validation, Investigation, Methodology, Writing - original draft, Project administration, Writing - review and editing

### Author ORCIDs

Go Kasuya (ID) http://orcid.org/0000-0003-1756-5764
Koichi Nakajo (ID) http://orcid.org/0000-0003-0766-7281

### Ethics

All experiments were approved by the Animal Care Committee of Jichi Medical University (Japan) under protocol no. 18027-03 and were performed according to guidelines.

### Decision letter and Author response

Decision letter https://doi.org/10.7554/eLife.81683.sa1
Author response https://doi.org/10.7554/eLife.81683.sa2

## Additional files

### Supplementary files

• Transparent reporting form

### Data availability

All data generated or analysed during this study are included in the manuscript and supporting file.

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
