## [Editor Report]

This study of physiologically important K^+^ channel complexes is expected to be important to electrophysiologists and biophysicists. This structure-motivated mutagenesis and biophysical study provide compelling evidence that a structural interface between a K^+^ channel and a class of modulatory subunits is a critical functional interface that determines the efficacy of the modulatory subunits.

---

## [Decision Letter]

**Decision letter after peer review:**

[Editors’ note: the authors submitted for reconsideration following the decision after peer review. What follows is the decision letter after the first round of review.]

Thank you for submitting your work entitled "Triad interaction stabilizes the voltage sensor domains in a constitutively open KCNQ1-KCNE3 channel" for consideration by *eLife*. Your article has been reviewed by 3 peer reviewers, including Jon T Sack as the Reviewing Editor and Reviewer #1, and the evaluation has been overseen by a Senior Editor.

Comments to the Authors:

We are sorry to say that, after consultation with the reviewers, we have decided that your work will not be considered further for publication by *eLife*.

The functional validation of the KCNQ1/KCNE3 structural interface is compelling, yet was not deemed to comprise a conceptual advance sufficient to merit publication in *eLife*. The reviewers agreed that validation of the interface by KCNQ1 F127A-KCNE3 G73L is an important result. However, this major conclusion could benefit from further experiments. Reviewers all thought that the data analysis needs substantial revision, and other claims should be tempered. If further experiments lead to additional mechanistic insights, the combined results could be submitted as a new manuscript.

*Reviewer #1 (Recommendations for the authors):*

This study aimed to identify amino acid residues on KCNE3 facing the S1 segment of KCNQ1 that are required for constitutive activity, and identify residues on the S4 segment which convert KCNE effects into voltage sensor movement. The research appears technically sound and expertly performed. The major strength of this work is the molecular redesign of KCNE3 and the S1 segment of KCNQ1 to rescue the functional interaction between KCNE3 and KCNQ1. However, this key result rests on a single pair of mutations. The weaknesses are some of the mechanistic interpretations pertaining to functional coupling between KCNQ1 and KCNE3. The manuscript achieved its aim of identifying amino acid residues on KCNE3 facing the S1 segment of KCNQ1 that are required for constitutive activity. This work also identifies mutations on the S4 segment which diminish KCNE1 effects, but data seems inadequate to support the claim that the residues identified on the S4 are the site of conversion of KCNE effects into voltage sensor movement, as alternate explanations also appear plausible. Importantly, this work presents strong evidence that the KCNE3/S1 interface observed structurally is accurate in its molecular detail, and crucial for inter-protein coupling. This study should cement the functional relevance of the KCNE3 interface with the S1 of KCNQ1 observed in a recent structure. Additionally. this work identifies key sites that perturb the modulation of KCNQ1 by KCNE proteins.

The KCNQ1 F127A-KCNE3 G73L result is compelling, it is strong conceptual evidence that the KCNE3/S1 interface forms as advertised. I think the paper would be better appreciated if it focused almost entirely on this result, and perhaps further investigate this key finding with a few more related experiments. The manuscript states that " Appropriate side-chain volumes at the interaction interface are necessary" yet could other physical/chemical concepts explain the reason for rescue?

Testing further sets of double mutations at positions 127 and 73 could test whether this apparent rescue by the swap of side chain bulk is more than just a happy coincidence, and whether bulk inversions at positions 127 and 73 could be found that produce channels even more functionally similar to WT KCNE/KCNQ

I wonder if this KCNQ1 F127A-KCNE3 G73L protein engineering could be used to knock in KCNE-resistant channels that will only respond when a mutant KCNE is expressed…possibly useful for experiments conceptually similar to DREADDs or Kevan Shokat's knob-in-hole chemical genetics.

line 134

"For the KCNQ1 F127A mutant, co-expression with KCNE3 F68A or V72A yielded G-V curves similar to that for KCNE3 WT, implying that F127 of the S1 segment and F68/V72 of KCNE3 are functionally coupled."

I disagree. these results just mean that all of the mutations eliminate the E3 effect. The functional coupling analysis here and in the following sentences (up to line 144) seems flawed, or maybe I am failing to understand it. What basis is there for "additive" shifting?

line 182

"Therefore, we concluded that M238 and V241 are the sites where the binding of KCNE3 is converted to the VSD movement's modulation."

I disagree, and suggest softening this conclusion. In combination with the structure, the possibility of this being a site of conversion does seems plausible, but these results do not make this certain. This result merely means that perturbation of these sites perturbs the coupling. This is consistent with being a site of physical coupling, or conversion, but based on this result alone, other physical possibilities also seem plausible. For example, this could be a site needed to align the site of conversion or There could be multiple sites of conversion.

Figure Supplement 1D

The confocal images do clearly show surface expression, but provide little information about the density of surface KCNE1, nor association with KCNQ1. Quantitation across multiple oocytes could strengthen this control.

Figure 3F, purple traces appear to be mislabeled…should be F130A +A69F?

table 1:

reporting the midpoints of the E1 mutations on their own (no KCNE), even if from another publication, would be helpful

G is conductance, but in table 1 current amplitudes are given (minor semantic issue)

Exclusion criteria for oocyte leak should be in the methods section

*Reviewer #2 (Recommendations for the authors):*

In this manuscript the authors inspected the cryo-EM structure of KCNQ1 and KCNE1 and made mutations to KCNQ1 and KCNE3 residues that appear to interact in the structure. These mutated residues are located in the transmembrane segment S1 and S4 of KCNQ1 and of KCNE3. The authors found that the KCNE4 mutations F68A, V72A and I 76 A abolished the ability of KCNE3 to shift GV relation to voltages more negative than -160 mV, which is similar to the findings of their previous study that the S1 mutations F127A and F130A also abolished the ability of KCNE3 to shift GV to negative voltages. A double mutation Q1 F127A + E3 G73L rescued a large part of the GV shift, supporting the observation from the structure that KCNQ1 S1 and KCNE3 may interact among these residues for the modulatory effects of KCNE3 on channel gating. The authors also measured VSD movements in the WT and these mutant channels using VCF, and concluded that the S1-E3 interactions at the S1 site are important for the KCNE3 induced shift of VSD activation to the negative voltages. The authors then found that mutations of S4, M238A and M241A, also reduced the ability of KCNE3 to shift the GV relation, and reduce VSD activation at negative voltages. Based on these results, the authors conclude that the interactions among S1, S4 and KCNE3 at the location of these residues modulate VSD activation and stabilize the open state of the channel. The manuscript presents some interesting data, including the Q1 F127A + E3 G73L results as evidence for S1-KCNE3 interaction. However, some of the conclusions made in the manuscript seem to be based on incomplete experiments or analyses of the data.

Some questions and comments are as follows.

1) Although mutant KCNE3 was shown to express in the surface membrane, it is not known if E3 association with Q1 is reduced by the mutations. It is known that Q1:E1 at different ratio can have different stoichiometry (as shown by the author of this manuscript) and the channels with different stoichiometry show different function including G-V and kinetics. Does KCNE3 share similar properties? Additional experiments may be needed to address this concern.

2) The authors used DF-160/DF60 to measure VSD activation. In VCF measurements it is not clear how VCF amplitude is related to VSD movements. This method seems to be flawed.

3) It is not clear why the authors did not analyze the VCF data to show FV. The relationship between FV and GV is not analyzed either. The authors made conclusions in reference of intermediate and activated states of the VSD, but these conclusions were not backed up by data of fluorescence or current measurements.

4) KCNE3 may also contact with S3 and S5 of KCNQ1 in the cryo-EM structure. Why did not the authors inspect the residues in these motifs? In addition, the cryo-EM structure shows the conformation of only one state, but the KCNE3-KCNQ1 interactions in other states are not known. These need to be considered in the structure motivated mutagenesis study.

5) Related to comment 4, how does this study reconcile with the studies by the McDonald lab (Melman et al. 2004 Neuron), which concluded that KCNE1 (and KCNE3) interacted with the pore domain of KCNQ1 to modulate channel activation?

6) Figure 2. What does "functionally coupled" mean? How is this defined? The interpretation of Figure 2 data is hard to follow and seems to be arbitrary. For example, for F127A the E3 mutations are "functionally coupled" when the GV is the same to E3 WT (Figure 2A & B), but for F130A, the E3 mutations are "functionally coupled" when the GV differs from E3 WT (Figure 2C & D).

7) Figure 4. The FV of the mutant channels + E3 WT shifts to negative voltages and shows two components. The shifts need to be characterized and compared to WT KCNQ1+E3. The ratio of DF-160/DF60 in Figure 4F does not reflect the shift and is misleading (see comment 2).

8) E1 differs from E3 in sequence and function but the S4 mutations are shown to affect E1 and E3 similarly, please explain. In addition, previous studies (Melman et al. 2001 JBC) showed that the triplet amino acids in the middle of the transmembrane segment of KCNE1 and 3 can switch the function between KCNE1 and KCNE3. How does this study reconcile with the previous results?

*Reviewer #3 (Recommendations for the authors):*

This is an interesting manuscript dealing with the functional effect of KCNE3 on KCNQ1 function. Using the latest KCNQ1/KCNE3 CryoEM structures, they test different residues for functional effects on S4 movement or gate opening. They identify some interactions between S1 and KCNE3 and that these interactions affect the voltage sensor S4 movement via two residues in S4. This is a timely study on a physiologically important K channel involved in many diseases. My main concern is how the analysis of the data was done. But if it stands up to more rigorous analysis, I think the findings are very important and would be interesting to a general audience.

1. Pg. 7. Line 108. Is ΔF-160mV/ΔF60mV a convincing measure for determining the voltage dependence? Can authors explain more about the parameter and why they choose this one? To me, it is not clear that this parameter will give unambiguous conclusions. For example, If F1 is shifted to more depolarized potentials, then the ratio will be bigger. But if F2 is shifted to the right instead, then the ratio will also be bigger. So how distinguish between these two cases with this parameter? Why didn't the authors use even more negative voltages (-180 mV) or positive voltages (+80 mV) to get complete FVs?

2. Also, normalizing the FVs at +60mV if they are not saturated at +60mV will distort the FVs. Not sure how to get around this problem though?

3. Pg. 9. Line 167. M238 and V241 are important for KCNE3 binding and modulation of KCNQ1, but no evidence shows that M238 and V241 interact with S1 of KCNQ1, even though they seem face to each other. In other words, the authors have not identified the functional coupling between S1 and S4. Therefore, the subtitle here is not appropriate.

4. Admittedly, KCNE3 interacts with S1 and S4, but the authors have not shown any evidence that S4 and S1 interact, and that these interactions are important for VSD modulation of KCNE1 or KCNE3. The title stating the triad interaction need to be revised. Any statement talking about triad interaction, such as line 73 and 278, should be avoided.

5. Pg. 8. Line 134-138. More explanation is needed for the functional couple between KCNQ1 and KCNE3. Why similar GV curves suggested they are functionally coupled? Why the additive GV curves suggested no interaction? Is this mutant cycle analysis?

6. Pg. 9. Line 151. I understand why the authors used mutation G73L, but wouldn't it be better to switch the two residues at F127-KNCQ1 and G73-KCNE3 (F127A-G73F) and see the role of Phe in the KCNQ1-KCNE3 interaction as they did with KCNQ1-F130A-A69F-KCNE3?

7. Pg. 11. Line 199. I wonder if the mutations affect the kinetics of the S4 movement? This would be helpful understanding how mutations in S4 affect the VSD of KCNQ1 by KCNE1 and KCNE3 modulation.

8. Figure 1D and 1F. Any evidence showing that KCNE3 is indeed co-expressed with KCNQ1? For me, some mutations, such as Q1-WT-E3-I76A, show very similar current and fluorescence traces with the WT Q1-E3. This concern is also raised for Q1-F127A-E3-G73L in Figure 3B and 3D. Some conclusions in the manuscript couldn't be made without showing KCNE3 or its mutations is present in the KCNQ1 channels.

9. Figure 1D and 1F, for the trace of -160 mV in Q1-WT-E3-WT, the fluorescence signal is lower at the tail voltage of -30 mV than at the resting voltage of -90 mV, which seems incorrect. I would expect S4 would move up higher during -30 mV than -90 mV, which means the fluorescence signal would be bigger at -30 mV than -90 mV. Please show better recordings or explain why. Please also check with Q1-F127A-E3-G73L in Figure 3D, Q1-M238A+E3 WT in Figure 4D.

10. Figure 5B. More positive voltages, at least +80 mV, should be used as in Figure 2D. The three GV curves are not even close to getting saturated at positive voltages. Please also provide GV at + 80 mV and +100 mV for Figure supplement 4B.

[Editors’ note: further revisions were suggested prior to acceptance, as described below.]

Thank you for resubmitting your work entitled "Optimized tight binding between the S1 segment and KCNE3 is required for the constitutively open nature of the KCNQ1-KCNE3 channel complex" for further consideration by *eLife*. Your revised article has been evaluated by Richard Aldrich (Senior Editor) and a Reviewing Editor.

The manuscript has been improved but there are some remaining issues that need to be addressed, as outlined below:

1. Provide data assessing whether similar interactions proposed for KCNE3 function are important for KCNE1 function. We imagine only a few key experiments will be needed and not an extensive study. (see Reviewer #3 comment 1)

2. Provide data testing whether the co-expression method allows a full association of KCNE3 with KCNQ1, by testing the dose-dependence of KCNE3 RNA for key KCNE3 mutations. (see Reviewer #3 comment 2)

3. Where fits of the FVs are not constrained enough, just report an upward bound for the V0.5. (see Reviewer #2 comment)

*Reviewer #1 (Recommendations for the authors):*

This manuscript is a transformed version of the one submitted prior. The new data and improved presentation provide very strong evidence that the KCNQ1 S1 – KCNE3 interface observed in a cryo-EM structure is critical for KCNE3 to exert its proper functional effect on KCNQ1. The volume scanning of each residue on both sides of a double mutant cycle analysis that resulted in rescue by multiple sets of interacting pairs makes it difficult to imagine that the binding interface observed in the structure is not an interface that enables KCNE3 function.

General suggestion:

The language in the paper refers to mutants "impairing" the modulation or being "tolerated". It seemed to me that the mutations may not impair the interaction so much as change the functional output of the interaction.

Can any conclusions be reached or compelling speculations made about whether the mutations prevent E3 from binding to all or a proportion of channels vs E3 binding well to all channels, but in a distorted conformation that induces a different GV?

It looked to me in Figures2L, 3G, 4N that bulkier residues consistently positively shifted the GV, suggesting the latter in those cases. I might speculate that the exact conformation of KCNE3, which is determined by the S1-E3 interface, determines the functional impact of E3 and that could at least partially explain the functional differences between KCNEs. Such speculation is not necessary but is my conceptual takeaway from this study.

Specific suggestions:

Line 112

" It seemed that the modulation depended on the side-chain volume: the more different the size was, the more significant was the impairment of modulation (Figure 2N). "

It could be helpful to quantitate what the side chain volumes are for G, A, V, L, F, W.

line 220

"which further strengthens the importance of the tight interaction between the S1 segment and KCNE3. "

I think it would be more appropriate to state that this is the strong evidence that the specific interaction between the S1 segment and KCNE3 is important. None of the single mutations really show that this precise interface is functionally critical.

line 265

"Overall, the results indicate that KCNE3 residues in the middle of the transmembrane segment are important for channel modulation, probably by preventing the S4 segment from going to the down position. "

Similarly I think it would be more appropriate to state that this is the strong evidence that it is an interaction between the S1 segment and KCNE3 that is important for modulating for VSD movement.

*Reviewer #2 (Recommendations for the authors):*

The authors have responded well to my critique and made several improvements of the manuscript. I now have only one comment that should be addressed:

The fits of the FVs are in some cases not constrained enough by the data, e.g. when the data do not tend to saturate at either the negative or positive range of voltages. In those cases, it is much better to just report an upward bound for the V0.5. For example, one can state that the V0.5>+60 mV for FVs that do tend to saturate at positive voltages and V0.5<-160 mV for FVs that do tend to saturate at negative voltages. The conclusions will still be the same and valid even without an exact V0.5 number, since for the single mutants you have a V0.5 that is different from +60mV or -160 mV.

*Reviewer #3 (Recommendations for the authors):*

In this interesting study the interaction between the S1 segment of KCNQ1 and the transmembrane segment of KCNE3 was shown to determine the shift of voltage dependence of activation of both the voltage sensor and the conductance. Guided by the KCNQ1-KCNE3 structure that was previously solved using cryo-EM, extensive mutations suggested that the S1-KCNE3 interaction depended on the size of the side chain of residues in these segments. Changing side chain sizes in either S1 or KCNE3 altered the conductance shifts, and at two pairs of S1-KCNE3 interaction residues, the complementary size changes rescued voltage shifts for both conductance and voltage sensor activation. The data are clearly presented with compelling conclusions. These results provide molecular mechanism for subunit modulation of a physiologically important ion channel. Some comments are as follows.

1. It is known that KCNE1, which is an important KCNQ1 regulatory subunit in the heart, also shifts voltage sensor activation to more negative voltages similarly as KCNE3. This study only focuses on KCNE3. While the study is well done with interesting results, the results would be more significant for publishing in *eLife* if some of the KCNQ1 mutations can be coexpressed with KCNE1 to examine if similar interactions proposed here are important for KCNE1 function. Such experiments do not have to be extensive as for the KCNE3 study. The insights can be very valuable. It seems that the previous version of the manuscript included some KCNE1 results. The authors should not cut out these results in the revised manuscript.

2. The authors addressed the comments from previous reviews on the issue of association of mutant KCNE3 with KCNQ1. 10 ng KCNQ1 was used to express the channel complex with 1 ng mutant KCNE3. To support that this method allows a full association of KCNE3 with KCNQ1, the functional dependence on different concentrations of KCNE3 RNA, a high and a low concentration, should be shown for at least a couple of KCNE3 mutations.

3. Line 98 and afterward: "∆G", should it be "G"? Please explain in the text that it is conductance.

[Editors’ note: further revisions were suggested prior to acceptance, as described below.]

Thank you for resubmitting your work entitled "Optimized tight binding between the S1 segment and KCNE3 is required for the constitutively open nature of the KCNQ1-KCNE3 channel complex" for further consideration by *eLife*. Your revised article has been evaluated by Richard Aldrich (Senior Editor) and a Reviewing Editor.

The manuscript has been improved but there are some remaining issues that need to be addressed, as outlined below:

Please include the KCNE1 results shown in the rebuttal in the manuscript, and address the mechanistic implications of these results. As reviewer #3 noted, the sequence of F127 mutations in KCNQ1 seem to shift the GV of both +KCNE1 and +KCNE3 in a similar way, even though KCNE1 and KCNE3 shift the GV of KCNQ1 in opposing fashions. This is very interesting, broadens the significance of the findings to the KCNE family generally, and deserves thoughtful mechanistic discussion. Also, please show data quantifying the reproducibility of the Cover Letter Figure 1D results show to back the claim that the F127A+E1 GV is shifted far in the positive direction.

*Reviewer #3 (Recommendations for the authors):*

The authors have responded to my comments. Yes, please include the KCNE1 results shown in the rebuttal in the manuscript, and a discussion of the comparison between KCNE1 and KCNE3 results will be significant since KCNE1 and KCNE3 modulate channel function with drastic differences. These results, on the other hand, do not show mechanistic differences. What would this comparison mean to the mechanism studied in this manuscript?

---

## [Author Response]

[Editors’ note: the authors resubmitted a revised version of the paper for consideration. What follows is the authors’ response to the first round of review.]

Comments to the Authors:We are sorry to say that, after consultation with the reviewers, we have decided that your work will not be considered further for publication by eLife.The functional validation of the KCNQ1/KCNE3 structural interface is compelling, yet was not deemed to comprise a conceptual advance sufficient to merit publication in eLife. The reviewers agreed that validation of the interface by KCNQ1 F127A-KCNE3 G73L is an important result. However, this major conclusion could benefit from further experiments. Reviewers all thought that the data analysis needs substantial revision, and other claims should be tempered. If further experiments lead to additional mechanistic insights, the combined results could be submitted as a new manuscript.Reviewer #1 (Recommendations for the authors):This study aimed to identify amino acid residues on KCNE3 facing the S1 segment of KCNQ1 that are required for constitutive activity, and identify residues on the S4 segment which convert KCNE effects into voltage sensor movement. The research appears technically sound and expertly performed. The major strength of this work is the molecular redesign of KCNE3 and the S1 segment of KCNQ1 to rescue the functional interaction between KCNE3 and KCNQ1. However, this key result rests on a single pair of mutations. The weaknesses are some of the mechanistic interpretations pertaining to functional coupling between KCNQ1 and KCNE3. The manuscript achieved its aim of identifying amino acid residues on KCNE3 facing the S1 segment of KCNQ1 that are required for constitutive activity. This work also identifies mutations on the S4 segment which diminish KCNE1 effects, but data seems inadequate to support the claim that the residues identified on the S4 are the site of conversion of KCNE effects into voltage sensor movement, as alternate explanations also appear plausible. Importantly, this work presents strong evidence that the KCNE3/S1 interface observed structurally is accurate in its molecular detail, and crucial for inter-protein coupling. This study should cement the functional relevance of the KCNE3 interface with the S1 of KCNQ1 observed in a recent structure. Additionally. this work identifies key sites that perturb the modulation of KCNQ1 by KCNE proteins.The KCNQ1 F127A-KCNE3 G73L result is compelling, it is strong conceptual evidence that the KCNE3/S1 interface forms as advertised. I think the paper would be better appreciated if it focused almost entirely on this result, and perhaps further investigate this key finding with a few more related experiments. The manuscript states that " Appropriate side-chain volumes at the interaction interface are necessary" yet could other physical/chemical concepts explain the reason for rescue?Testing further sets of double mutations at positions 127 and 73 could test whether this apparent rescue by the swap of side chain bulk is more than just a happy coincidence, and whether bulk inversions at positions 127 and 73 could be found that produce channels even more functionally similar to WT KCNE/KCNQI wonder if this KCNQ1 F127A-KCNE3 G73L protein engineering could be used to knock in KCNE-resistant channels that will only respond when a mutant KCNE is expressed…possibly useful for experiments conceptually similar to DREADDs or Kevan Shokat's knob-in-hole chemical genetics.

We greatly appreciate your valuable comments. In this revised manuscript, we exclusively focused on the interaction between the S1 and KCNE3, following your suggestions. We further investigated the effect of mutations on the interaction face between the S1 and KCNE3 and successfully identified at least another pair (S57 and I145).

The idea of knock-in KCNE-resistant channels sounds cool. That might work!

line 134"For the KCNQ1 F127A mutant, co-expression with KCNE3 F68A or V72A yielded G-V curves similar to that for KCNE3 WT, implying that F127 of the S1 segment and F68/V72 of KCNE3 are functionally coupled."I disagree. these results just mean that all of the mutations eliminate the E3 effect. The functional coupling analysis here and in the following sentences (up to line 144) seems flawed, or maybe I am failing to understand it. What basis is there for "additive" shifting?

As you pointed out, the analysis may be flawed, and we abandoned this part in this revision. Therefore, the sentence no longer exists.

line 182"Therefore, we concluded that M238 and V241 are the sites where the binding of KCNE3 is converted to the VSD movement's modulation."I disagree, and suggest softening this conclusion. In combination with the structure, the possibility of this being a site of conversion does seems plausible, but these results do not make this certain. This result merely means that perturbation of these sites perturbs the coupling. This is consistent with being a site of physical coupling, or conversion, but based on this result alone, other physical possibilities also seem plausible. For example, this could be a site needed to align the site of conversion or There could be multiple sites of conversion.

As we focused on the S1 segment and abandoned the S4 analysis, this sentence no longer exists.

Figure Supplement 1DThe confocal images do clearly show surface expression, but provide little information about the density of surface KCNE1, nor association with KCNQ1. Quantitation across multiple oocytes could strengthen this control.

We agree with your comments. The confocal images only show the surface expression and do not provide any information on the association with KCNQ1. On the other hand, as seen in Figure 3 and Figure supplements 9-12, all but I76A mutant showed substantial changes in voltage dependence. Furthermore, as seen in Figure supplement 8, we confirmed that our experiment condition where 10 ng RNA of KCNQ1 co-injected with 1 ng RNA of KCNE3 is sufficient to fully modulate KCNQ1 currents at least in the cases of KCNE3 WT and two representative mutants (S57A and G73L). Therefore, we believe most KCNE3 mutants bind and modulate KCNQ1 channels.

Figure 3F, purple traces appear to be mislabeled…should be F130A +A69F?

This figure no longer exists.

Table 1: reporting the midpoints of the E1 mutations on their own (no KCNE), even if from another publication, would be helpful G is conductance, but in table 1 current amplitudes are given (minor semantic issue)

This revised manuscript reports the midpoints and other parameters for all KCNQ1 mutations (with or without KCNE3) and KCNE3 mutants (Tables supplement 1-3).

As you pointed out, G should have been I. We have corrected them in the Table supplements.

Exclusion criteria for oocyte leak should be in the methods section

We only used oocytes with a holding current less than -0.2 uA at -90 mV in TEVC and -0.3 µA at -100 mV in VCF. We have added the criteria in the methods section.

Reviewer #2 (Recommendations for the authors):In this manuscript the authors inspected the cryo-EM structure of KCNQ1 and KCNE1 and made mutations to KCNQ1 and KCNE3 residues that appear to interact in the structure. These mutated residues are located in the transmembrane segment S1 and S4 of KCNQ1 and of KCNE3. The authors found that the KCNE4 mutations F68A, V72A and I 76 A abolished the ability of KCNE3 to shift GV relation to voltages more negative than -160 mV, which is similar to the findings of their previous study that the S1 mutations F127A and F130A also abolished the ability of KCNE3 to shift GV to negative voltages. A double mutation Q1 F127A + E3 G73L rescued a large part of the GV shift, supporting the observation from the structure that KCNQ1 S1 and KCNE3 may interact among these residues for the modulatory effects of KCNE3 on channel gating. The authors also measured VSD movements in the WT and these mutant channels using VCF, and concluded that the S1-E3 interactions at the S1 site are important for the KCNE3 induced shift of VSD activation to the negative voltages. The authors then found that mutations of S4, M238A and M241A, also reduced the ability of KCNE3 to shift the GV relation, and reduce VSD activation at negative voltages. Based on these results, the authors conclude that the interactions among S1, S4 and KCNE3 at the location of these residues modulate VSD activation and stabilize the open state of the channel. The manuscript presents some interesting data, including the Q1 F127A + E3 G73L results as evidence for S1-KCNE3 interaction. However, some of the conclusions made in the manuscript seem to be based on incomplete experiments or analyses of the data.Some questions and comments are as follows.1) Although mutant KCNE3 was shown to express in the surface membrane, it is not known if E3 association with Q1 is reduced by the mutations. It is known that Q1:E1 at different ratio can have different stoichiometry (as shown by the author of this manuscript) and the channels with different stoichiometry show different function including G-V and kinetics. Does KCNE3 share similar properties? Additional experiments may be needed to address this concern.

Thank you for raising this critical point. We also think the stoichiometry could be various, especially when E3 expression is low, as in the case of Q1:E1. To verify the E3 expression level was high enough, we conducted KCNQ1-KCNE3 current measurements with various amounts of KCNE3 RNAs (Figure supplement 8). According to the experiments, 1 ng of KCNE3 RNA is sufficient to fully modulate 10 ng of KCNQ1 channels. We added these descriptions to explain our rational experiment design (Lines 146-148).

2) The authors used DF-160/DF60 to measure VSD activation. In VCF measurements it is not clear how VCF amplitude is related to VSD movements. This method seems to be flawed.

Thank you for pointing the problem out. We admit the DF-160/DF60 index may not be appropriate and is no longer used for evaluation.

3) It is not clear why the authors did not analyze the VCF data to show FV. The relationship between FV and GV is not analyzed either. The authors made conclusions in reference of intermediate and activated states of the VSD, but these conclusions were not backed up by data of fluorescence or current measurements.

We have added G-Vs and F-Vs for all VCF data in Figure 5.

4) KCNE3 may also contact with S3 and S5 of KCNQ1 in the cryo-EM structure. Why did not the authors inspect the residues in these motifs? In addition, the cryo-EM structure shows the conformation of only one state, but the KCNE3-KCNQ1 interactions in other states are not known. These need to be considered in the structure motivated mutagenesis study.

As you pointed out, other segments are in contact with KCNE3 in the cryo-EM structure. In this paper, however, we focused on the S1 segment in the VSD in this study. Of course, our results do not exclude any significance of the contact with the pore domain (S5 and S6).

We understand that the cryo-EM structure represents one of many states. Still, our mutagenesis studies clearly show that most of the interactions between S1 and KCNE3 in the structure are required for the constitutive activity of the KCNQ1-KCNE3 channel.

5) Related to comment 4, how does this study reconcile with the studies by the McDonald lab (Melman et al. 2004 Neuron), which concluded that KCNE1 (and KCNE3) interacted with the pore domain of KCNQ1 to modulate channel activation?

As answered for comment 4, our results do not exclude any significance of the contact with the pore domain (S5 and S6). However, according to Barro-Soria et al. (PNAS, 2017), KCNE1 affects both the S4 movement and the gate, while KCNE3 affects the S4 movement and only affects the gate indirectly. Therefore, the interaction between the pore domain and KCNE3 could be less important than in the case of KCNE1.

6) Figure 2. What does "functionally coupled" mean? How is this defined? The interpretation of Figure 2 data is hard to follow and seems to be arbitrary. For example, for F127A the E3 mutations are "functionally coupled" when the GV is the same to E3 WT (Figure 2A & B), but for F130A, the E3 mutations are "functionally coupled" when the GV differs from E3 WT (Figure 2C & D).

As we answered to reviewer #1, the analysis may be flawed, and we abandoned this part in this revision. Therefore, the sentence no longer exists.

7) Figure 4. The FV of the mutant channels + E3 WT shifts to negative voltages and shows two components. The shifts need to be characterized and compared to WT KCNQ1+E3. The ratio of DF-160/DF60 in Figure 4F does not reflect the shift and is misleading (see comment 2).

Thank you for the comment. We updated the FV analysis to clearly show the two components in the Q1WT + E3 WT. Mutations on KCNQ1 diminished the first component, suggesting that the intermediate state of the VSD became harder to maintain by the mutations. Then, E3 mutants partly restored the first component (Figure 5).

8) E1 differs from E3 in sequence and function but the S4 mutations are shown to affect E1 and E3 similarly, please explain. In addition, previous studies (Melman et al. 2001 JBC) showed that the triplet amino acids in the middle of the transmembrane segment of KCNE1 and 3 can switch the function between KCNE1 and KCNE3. How does this study reconcile with the previous results?

In this manuscript, we no longer examined KCNE1 and solely focused on KCNE3. However, the functional difference between KCNE1 and KCNE3 is still an open and vital question that remains to be seen. One possible explanation is the intermediate state of KCNQ1-KCNE1 is non-conductive while the intermediate state of KCNQ1-KCNE3 is conductive. Amino acid sequence difference may contribute to the conductive state of the intermediate state.

Reviewer #3 (Recommendations for the authors):This is an interesting manuscript dealing with the functional effect of KCNE3 on KCNQ1 function. Using the latest KCNQ1/KCNE3 CryoEM structures, they test different residues for functional effects on S4 movement or gate opening. They identify some interactions between S1 and KCNE3 and that these interactions affect the voltage sensor S4 movement via two residues in S4. This is a timely study on a physiologically important K channel involved in many diseases. My main concern is how the analysis of the data was done. But if it stands up to more rigorous analysis, I think the findings are very important and would be interesting to a general audience.1. Pg. 7. Line 108. Is ΔF-160mV/ΔF60mV a convincing measure for determining the voltage dependence? Can authors explain more about the parameter and why they choose this one? To me, it is not clear that this parameter will give unambiguous conclusions. For example, If F1 is shifted to more depolarized potentials, then the ratio will be bigger. But if F2 is shifted to the right instead, then the ratio will also be bigger. So how distinguish between these two cases with this parameter? Why didn't the authors use even more negative voltages (-180 mV) or positive voltages (+80 mV) to get complete FVs?

As we answered reviewer #2’s comment, we admitΔF-160mV/ΔF60mV may not be appropriate. We abandoned ΔF-160mV/ΔF60mV. Instead, we used the complete F‑V curves as suggested (Figure 5).

2. Also, normalizing the FVs at +60mV if they are not saturated at +60mV will distort the FVs. Not sure how to get around this problem though?

We presented full F-V curves and fitted them with the Boltzmann equation (Figure 5). F-Vs are normalized accordingly.

3. Pg. 9. Line 167. M238 and V241 are important for KCNE3 binding and modulation of KCNQ1, but no evidence shows that M238 and V241 interact with S1 of KCNQ1, even though they seem face to each other. In other words, the authors have not identified the functional coupling between S1 and S4. Therefore, the subtitle here is not appropriate.

We focused on the S1 segment in this manuscript and abandoned the S4 data.

4. Admittedly, KCNE3 interacts with S1 and S4, but the authors have not shown any evidence that S4 and S1 interact, and that these interactions are important for VSD modulation of KCNE1 or KCNE3. The title stating the triad interaction need to be revised. Any statement talking about triad interaction, such as line 73 and 278, should be avoided.

Because we no longer examined the triad interaction, we have changed the title accordingly.

5. Pg. 8. Line 134-138. More explanation is needed for the functional couple between KCNQ1 and KCNE3. Why similar GV curves suggested they are functionally coupled? Why the additive GV curves suggested no interaction? Is this mutant cycle analysis?

As we answered reviewers #1 and #2 comments, the “functional coupling” analysis may be flawed, and we abandoned this part in this revision. Therefore, the sentence no longer exists.

6. Pg. 9. Line 151. I understand why the authors used mutation G73L, but wouldn't it be better to switch the two residues at F127-KNCQ1 and G73-KCNE3 (F127A-G73F) and see the role of Phe in the KCNQ1-KCNE3 interaction as they did with KCNQ1-F130A-A69F-KCNE3?

Thank you for the comment. In this version, we examined five residues, including G73L, and found that G73L was the best option to restore the F127A mutant (Figure 4A-G). Therefore, Phe or its ring structure may not be required here.

7. Pg. 11. Line 199. I wonder if the mutations affect the kinetics of the S4 movement? This would be helpful understanding how mutations in S4 affect the VSD of KCNQ1 by KCNE1 and KCNE3 modulation.

Because we abandoned the S4 experiments, this part no longer exists.

8. Figure 1D and 1F. Any evidence showing that KCNE3 is indeed co-expressed with KCNQ1? For me, some mutations, such as Q1-WT-E3-I76A, show very similar current and fluorescence traces with the WT Q1-E3. This concern is also raised for Q1-F127A-E3-G73L in Figure 3B and 3D. Some conclusions in the manuscript couldn't be made without showing KCNE3 or its mutations is present in the KCNQ1 channels.

Thank you for raising this critical point. It is a little difficult to prove the KCNE3 indeed co-expressed with KCNQ1. As we answered to reviewer #2’s comment, to verify the E3 expression level was high enough, we conducted KCNQ1-KCNE3 current measurements with various amounts of KCNE3 RNAs (Figure supplement 8). According to the experiments, 1 ng of KCNE3 RNA is high enough to fully modulate KCNQ1 channels. The G73L and S57A showed similar tendencies. Most KCNE3 mutants show different z and V1/2 values, as shown in Table supplement 2, indicating they successfully modulated KCNE3. Only one exception is I76A. KCNQ1+KCNE3 I76A showed similar z and V1/2 values to KCNQ1 WT alone (Table supplements 1-2, Figure supplement 12). It may fail to bind KCNQ1.

9. Figure 1D and 1F, for the trace of -160 mV in Q1-WT-E3-WT, the fluorescence signal is lower at the tail voltage of -30 mV than at the resting voltage of -90 mV, which seems incorrect. I would expect S4 would move up higher during -30 mV than -90 mV, which means the fluorescence signal would be bigger at -30 mV than -90 mV. Please show better recordings or explain why. Please also check with Q1-F127A-E3-G73L in Figure 3D, Q1-M238A+E3 WT in Figure 4D.10. Figure 5B. More positive voltages, at least +80 mV, should be used as in Figure 2D. The three GV curves are not even close to getting saturated at positive voltages. Please also provide GV at + 80 mV and +100 mV for Figure supplement 4B.

We apologize for the errors and the confusion in the previous version. We do not know why this happened. It could be overcompensation of fluorescence bleaching. We abandoned the data in this version and carefully remeasured the F-V relationships in Figure 5.

As large endogenous currents appear over +80 mV, we only applied membrane potential up to +60 mV.

[Editors’ note: further revisions were suggested prior to acceptance, as described below.]

The manuscript has been improved but there are some remaining issues that need to be addressed, as outlined below:Reviewer #1 (Recommendations for the authors):This manuscript is a transformed version of the one submitted prior. The new data and improved presentation provide very strong evidence that the KCNQ1 S1 – KCNE3 interface observed in a cryo-EM structure is critical for KCNE3 to exert its proper functional effect on KCNQ1. The volume scanning of each residue on both sides of a double mutant cycle analysis that resulted in rescue by multiple sets of interacting pairs makes it difficult to imagine that the binding interface observed in the structure is not an interface that enables KCNE3 function.General suggestion:The language in the paper refers to mutants "impairing" the modulation or being "tolerated". It seemed to me that the mutations may not impair the interaction so much as change the functional output of the interaction.Can any conclusions be reached or compelling speculations made about whether the mutations prevent E3 from binding to all or a proportion of channels vs E3 binding well to all channels, but in a distorted conformation that induces a different GV?It looked to me in Figures2L, 3G, 4N that bulkier residues consistently positively shifted the GV, suggesting the latter in those cases. I might speculate that the exact conformation of KCNE3, which is determined by the S1-E3 interface, determines the functional impact of E3 and that could at least partially explain the functional differences between KCNEs. Such speculation is not necessary but is my conceptual takeaway from this study.

We agree the reviewer #1’s insightful speculation. As shown in Figure supplement 8G-R, no matter how much the KCNE3 RNA ratio increased, the modulation was not fully recovered in KCNE3 mutants. This result indicates the latter is likely the case. Accordingly, as reviewer #1 suggested, we stopped using the word “impairing” and “tolerated” and changed them like “changed the functional output” or “distorted” (lines 90-91, 114-115, 133-134, 138, 141, 163-164, 167-168, 172-173, 181-182, 184, 190-191, 195, 201, 207, and 232, highlighted in yellow).

Specific suggestions:Line 112" It seemed that the modulation depended on the side-chain volume: the more different the size was, the more significant was the impairment of modulation (Figure 2N). "It could be helpful to quantitate what the side chain volumes are for G, A, V, L, F, W.

According to this comment, we added the volumes of amino acid residues in Figure 1D.

line 220"which further strengthens the importance of the tight interaction between the S1 segment and KCNE3. "I think it would be more appropriate to state that this is the strong evidence that the specific interaction between the S1 segment and KCNE3 is important. None of the single mutations really show that this precise interface is functionally critical.

According to this comment, we changed the corresponding sentence in lines 225-226.

line 265"Overall, the results indicate that KCNE3 residues in the middle of the transmembrane segment are important for channel modulation, probably by preventing the S4 segment from going to the down position. "Similarly I think it would be more appropriate to state that this is the strong evidence that it is an interaction between the S1 segment and KCNE3 that is important for modulating for VSD movement.

According to this comment, we changed the corresponding sentence in lines 270-272.

Reviewer #2 (Recommendations for the authors):The authors have responded well to my critique and made several improvements of the manuscript. I now have only one comment that should be addressed:The fits of the FVs are in some cases not constrained enough by the data, e.g. when the data do not tend to saturate at either the negative or positive range of voltages. In those cases, it is much better to just report an upward bound for the V0.5. For example, one can state that the V0.5>+60 mV for FVs that do tend to saturate at positive voltages and V0.5<-160 mV for FVs that do tend to saturate at negative voltages. The conclusions will still be the same and valid even without an exact V0.5 number, since for the single mutants you have a V0.5 that is different from +60mV or -160 mV.

We are pleased that we were able to respond to your previous comments appropriately. According to the additional comment, we revised the table (Figure 5-source data 2) as well as lines 259 and 268.

Reviewer #3 (Recommendations for the authors):In this interesting study the interaction between the S1 segment of KCNQ1 and the transmembrane segment of KCNE3 was shown to determine the shift of voltage dependence of activation of both the voltage sensor and the conductance. Guided by the KCNQ1-KCNE3 structure that was previously solved using cryo-EM, extensive mutations suggested that the S1-KCNE3 interaction depended on the size of the side chain of residues in these segments. Changing side chain sizes in either S1 or KCNE3 altered the conductance shifts, and at two pairs of S1-KCNE3 interaction residues, the complementary size changes rescued voltage shifts for both conductance and voltage sensor activation. The data are clearly presented with compelling conclusions. These results provide molecular mechanism for subunit modulation of a physiologically important ion channel. Some comments are as follows.

We appreciate your valuable comments. The responses to each comment are as follows.

1. It is known that KCNE1, which is an important KCNQ1 regulatory subunit in the heart, also shifts voltage sensor activation to more negative voltages similarly as KCNE3. This study only focuses on KCNE3. While the study is well done with interesting results, the results would be more significant for publishing in eLife if some of the KCNQ1 mutations can be coexpressed with KCNE1 to examine if similar interactions proposed here are important for KCNE1 function. Such experiments do not have to be extensive as for the KCNE3 study. The insights can be very valuable. It seems that the previous version of the manuscript included some KCNE1 results. The authors should not cut out these results in the revised manuscript.

Thank you for this constructive comment. Following this comment, we performed a preliminary mutational analysis of KCNQ1 F127 mutants (F127A, F127V, F127L, and F127W) to investigate the functional relationship between the S1 segment and KCNE1 (Author response image 1). We chose the F127 residue here since its modulation by KCNE3 depends on its side-chain volume size (Figure 2A-G, N), and the KCNQ1 F127A mutant-KCNE3 G73L mutant pair restored the modulation of KCNQ1 by KCNE3 (Figure 4A-G). All the F127 mutants potentiated KCNE1-dependent half-activation voltage (*V_1/2_*) shifts. Moreover, the more different side chain sizes induced the more significant *V_1/2_* shifts. (Please note that the G-V curve for F127A+E1 is omitted in the Cover letter figure 1K because the curve is shifted far in the positive direction and is impossible to fit with the Boltzmann function.) These results suggest that, at least for the F127 residue, the modulation by KCNE1 depended on the side-chain volume, as in the case of KCNE3. However, we would like to keep the KCNE1 results for the next manuscript because we think it is necessary to show all the scanning results of S1 residue to claim possible side-chain volume dependence on the KCNE1 modulation. (We hope the next one will lead to an exciting result that can suffice to submit *eLife*.) Adding the partial F127 residue result to this manuscript may confuse readers. We would still be happy to add the KCNE1 results to the current manuscript if reviewer #3 thinks it is indispensable.

**Author response image 1. sa2fig1:** Functional effects of KCNQ1 F127 mutants on KCNQ1 modulation by KCNE1. (A-N) Representative current traces (A-J) and G-V relationships (K-N) of KCNQ1 WT and F127 mutants with or without KCNE1 WT. (O) The half-activation voltage of KCNQ1 WT and F127 mutants with (filled bars) or without (open bars) KCNE1 WT. > 60 means over 60 mV. Error bars indicate ± s.e.m. for *n* = 5 in (I-M). ** and *** indicate **P < 0.01 and ***P < 0.001.

2. The authors addressed the comments from previous reviews on the issue of association of mutant KCNE3 with KCNQ1. 10 ng KCNQ1 was used to express the channel complex with 1 ng mutant KCNE3. To support that this method allows a full association of KCNE3 with KCNQ1, the functional dependence on different concentrations of KCNE3 RNA, a high and a low concentration, should be shown for at least a couple of KCNE3 mutations.

We have already presented the suggested experiment for KCNE3 S57A and G73L mutants in Figure 3—figure supplement 1G-R. We chose the KCNE3 mutants since they are functionally coupling with KCNQ1 I145F and F127A mutants, respectively (Figure 4). In both KCNE3 mutants, 0.3 ng KCNE3 RNA is enough to achieve maximum modulation with 10 ng KCNQ1 WT RNA. These results support that our expression condition through the experiment (10 ng KCNQ1 RNA and 1 ng KCNE3 RNA) is reasonable.

3. Line 98 and afterward: "∆G", should it be "G"? Please explain in the text that it is conductance.

Thank you for the suggestion, and we are sorry for our misconversion. DG in line 98 and afterward is DG (δ-G). We used this parameter to show the relative conductance at -100 mV and explained it in lines 103-105. In the revised version, we corrected the misconversion.

[Editors’ note: further revisions were suggested prior to acceptance, as described below.]

The manuscript has been improved but there are some remaining issues that need to be addressed, as outlined below:Please include the KCNE1 results shown in the rebuttal in the manuscript, and address the mechanistic implications of these results. As reviewer #3 noted, the sequence of F127 mutations in KCNQ1 seem to shift the GV of both +KCNE1 and +KCNE3 in a similar way, even though KCNE1 and KCNE3 shift the GV of KCNQ1 in opposing fashions. This is very interesting, broadens the significance of the findings to the KCNE family generally, and deserves thoughtful mechanistic discussion. Also, please show data quantifying the reproducibility of the Cover Letter Figure 1D results show to back the claim that the F127A+E1 GV is shifted far in the positive direction.Reviewer #3 (Recommendations for the authors):The authors have responded to my comments. Yes, please include the KCNE1 results shown in the rebuttal in the manuscript, and a discussion of the comparison between KCNE1 and KCNE3 results will be significant since KCNE1 and KCNE3 modulate channel function with drastic differences. These results, on the other hand, do not show mechanistic differences. What would this comparison mean to the mechanism studied in this manuscript?

We appreciate the reviewer’s constructive comments. We have added the new Figure 6 of KCNQ1 F127 mutants co-expressed with KCNE1 WT. We have also added the Results section (lines 274-286) and the Discussion section (lines 350-361). As discussed in the manuscript, we hypothesize the S1 segment may have a similar role in the KCNE1 and KCNE3 modulations, which is to help stabilize the intermediate state of the voltage sensor domain. Further extensive study is required for the possible role of the S1 segment in the KCNE1 modulation.